# Multiomics approaches disclose very-early molecular and cellular switches during insect-venom allergen-specific immunotherapy: an observational study

Allergen-specific immunotherapy (AIT) induces immune tolerance, showing the highest success rate (>95%) for insect venom while a much lower chance for pollen allergy. However, the molecular switches leading to successful durable tolerance restoration remain elusive. The primary outcome of this observational study is the comprehensive immunological cellular characterization during the AIT initiation phase, whereas the secondary outcomes are the serological and Th2-cell-type-specific transcriptomic analyses. Here we apply a multilayer-omics approach to reveal dynamic peripheral immune landscapes during the AIT-initiation phase in venom allergy patients (VAP) versus pollen-allergic and healthy controls. Already at baseline, VAP exhibit altered abundances of several cell types, including classical monocytes (cMono), CD4[+] hybrid type 1-type 17 cells (Th1-Th17 or Th1/17) and CD8[+] counterparts (Tc1-Tc17 or Tc1/17). At 8-24 h following AIT launch in VAP, we identify a uniform AIT-elicited pulse of late-transitional/IL-10-producing B cells, IL-6 signaling within Th2 cells and non-inflammatory serum-IL-6 levels. Sequential induction of activation and survival protein markers also immediately occur. A disequilibrium between serum IL-6 and cMono in VAP baseline is restored at day seven following AIT launch. Our longitudinal analysis discovers molecular switches during initiation-phase insect-venom AIT that secure long-term outcomes. Trial number: NCT02931955.

Repeated administration of relevant specific allergens, also known as allergen-specific immunotherapy (AIT), to patients with IgE-mediated allergy is the only curative option available. Notably, although AIT has been applied to many types of allergic diseases, the rates of clinical cure with a complete control of the allergic reaction in anaphylactic patients[1] in case of re-exposures were the highest (>95%) after the accomplishment of insect-venom AIT[2,3]. This fact makes insect-venom AIT a prime candidate as a clinical model to investigate the immune factors underpinning the very-high long-term curative rates.

AIT should be performed for at least three years to ensure a maximum immune tolerance-inducing effect[4,5]. Most of the studies have so far focused on the investigation of specific immune changes that occur later during the long-lasting AIT or after AIT completion. It has been well documented that AIT induces a decrease in various soluble and cellular allergy effectors. This is demonstrated by a reduction in IgE levels[6,7], a decrease in the activation and degranulation of mast cells and basophils[8,9], and lower frequencies of allergen-specific T helper 2 (Th2) cells[10,11] and type 2 innate lymphoid cells (ILC2)[12]. In the meantime, the allergic-response dampening factors or

e-mail: Feng.Hefeng@lih.lu; Markus.Ollert@lih.lu

cell types, such as IgE-blocking antibodies (Abs), i.e., mainly allergen-specific IgG4[13] and the immune regulatory subsets with suppressive functions [i.e., CD4+ regulatory T cells (Tregs) and B cells (Bregs)] are enhanced by AIT in the long run[8,14–19]. Nevertheless, AIT takes at least three years, requiring a long-term compliance of patients and being resource demanding.

Therefore, it is invaluable to investigate whether and which robust early-window cellular or molecular predictive events could aid in the decision of initiating and continuing AIT[20] as early as possible. In this work, in order to identify the early molecular and cellular switches from pathological type-2 (T2) immunity[21,22] to long-term immunological tolerance induced by insect-venom AIT, we compare dynamic immune responses between VAP and pollen allergy patients (PAP) already within the first hours and days following AIT launch, as AIT for the latter is known to exhibit a much lower likelihood of durable success. Furthermore, considering the effects of diurnal fluctuation (i.e., circadian rhythm) in immune cell circulation and function[23], here we stringently control for the exact blood sampling hours at each visit of each participant. To this end, our integrative analysis of the early dynamic immune landscape provides a unique resource by deeply characterizing ~200 million of peripheral immune cells and ~200 samples of purified Th2 cells collected in the initiation phase (mainly build-up phase including baseline) of AIT in a real-world, prospective and longitudinal cohort. Such a patient-based, ex-vivo, hypothesis-free analysis will allow to generate unexpected testable cellular and molecular hypotheses for identifying early predictive biomarkers for long-term AIT outcomes and targets for enhancing AIT efficacy in T2 diseases, beyond insect venom anaphylaxis.

## Results

### Revealing disturbed CD4+ and CD8+ T-cell subsets during the initiation phase of AIT

Here we administered AIT in two groups of patients allergic to insect venom or pollen. For 18 VAP, we implemented an ultra-rush (within eight hours) up-dosing protocol while a conventional (within six weeks) up-dosing AIT was administered in 16 PAP (Fig. 1a, Methods). As neither at the start of the project nor subsequently comparable ultra-rush up-dosing AIT products were licensed for pollen allergy, the exact sampling periods differed between VAP and PAP. Thus, our sampling schemes were critically chosen to cover comparable phases during the build-up of AIT until the targeted maintenance dose of allergens was achieved for both types of AIT. We performed time-series sampling of peripheral blood during the first seven days, including baseline before AIT, in VAP. As for PAP, we extended our sampling period to two weeks (2 W), six weeks (6 W) and 12 weeks (12 W). To benchmark with the natural immune fluctuations, we also recruited 10 healthy controls (HC), who did not receive any AIT or other immunotherapy, and sampled seven times within 12 weeks to match various time points of both patient groups. For the samples collected at each time point, we performed a multi-layer cellular and molecular omics (multi-omics) analysis, including single-cell mass cytometry (CyTOF) for systematic deep immunophenotyping[24–30] of 77 immune subsets, sorting-enabled Th2-cell-type-specific and whole-genome RNA sequencing (RNA-seq), kinome analysis of ~350 kinases, multiplexing analysis of 10 serological cytokines and conventional 17-parameter whole-blood-count analysis (Fig. 1b). Our multi-omics hypothesis-free analysis was accompanied by a detailed clinical workup in all the patients (Supplementary Table 1). For details of various immunological datasets from each participant at each time point, please visit our interactive resource website that allows investigators to examine all the datasets and figures in an interactive and interlinked manner (Fig. 1c, Methods).

We first integrated and compared the datasets of single-cell CyTOF analysis (for the gating strategy and marker information, refer to Supplementary Fig. 1, Supplementary Tables 2 and 3), serological cytokine multiplexing assays and whole-blood-count analysis between

VAP and HC at various matched time points. Already at baseline, and at the three following very-early time points [8 h, 24 h and day 7 (D7)], VAP demonstrated general immunological features that were different from HC in the principal component analysis (PCA) (Fig. 2a, Supplementary Fig. 2A). Meanwhile, the resulting peripheral immune responses among VAP induced by AIT were higher than the natural fluctuation of immune cells or soluble factors in HC over the same period (Supplementary Fig. 2B).

We next asked which specific immune subsets exhibited differential frequencies between VAP and HC at various matched time points. To systematically identify the changed subsets, we used the volcano plot to visualize (Fig. 2b) the results of all the 77 immune cell subsets defined by the combinations of various markers (Supplementary Fig. 1, Supplementary Tables 2 and 3 as well as the i3Dare website). At least 10 different immune subsets were significantly higher or lower in relative frequency (percentage) among total living singlets (refer to Methods) in VAP vs. HC, although not all of them showed a significant difference at each time point (Fig. 2b). Importantly, Th2 cells (CD45RO+CD45RA−CXCR3−CCR6−CCR4+, for the full gating strategy, please refer to Supplementary Fig. 1) showed a substantially higher level in VAP vs. HC at all time points. As allergy is known to be a predominantly IgE-driven Th2-mediated pathology[31–34], this observation was expected. Simultaneously, the pathogenic allergen-specific Th2 cells, characterized as CRTH2+ Th2 cells (CD3+CD56−TCRgd−CD4+CD8−CD45RO+CD45RA−CXCR3−CCR6−CCR4+C-CRTH2+) based on the markers established by the previous works[11,35,36], were also substantially more abundant in VAP (Figs. 2b, 3a, b). A dominant response of total and pathogenic Th2 cells in VAP revealed by our supervised CyTOF analysis was further consolidated by an unsupervised clustering analysis[37] (Fig. 3c). In contrast to the scenario in VAP, total CD4+ Th2 cells were not significantly changed in PAP vs. HC at any of the tested time points during the first 12 weeks (Supplementary Fig. 2C). The general immune features of PAP started to show a trend to be different from that of HC only at 12 W (Supplementary Fig. 2D). Indeed, PCA showed that at both matched early time points (baseline and 24 h), VAP and PAP exhibited distinguishable peripheral immune fingerprints (Supplementary Fig. 2E, F).

Notably, although CD8+ T cells have been only occasionally described in patient cohorts or animal studies in the allergy fields[38–41], the percentage of CCR4+CXCR3− type-2 cytotoxic CD8+ T cells (CD8+ Tc2, Fig. 3d, e)[42] among living cells was also higher in VAP vs. HC at each time point including baseline, while the percentage of CCR4−CXCR3+ type-1 CD8+ T cells (CD8+ Tc1) was modestly but significantly decreased (Supplementary Fig. 3A). CD8+ Tc2-like cells, referred to as the direct parent gate of CD8+ Tc2 (Supplementary Fig. 1), were also among the two significantly higher immune subsets in PAP vs. HC at several time points (Supplementary Figs. 2b, 3b). Remarkably, the percentage of CD45RA+CD8+ regulatory T cells (Tregs) (CD8+CD25highCD127lowCD45RA+) was significantly lower in VAP vs. HC at both baseline and D7 (Figs. 2b and 3f). A lower percentage of CD45RA+ and total CD8+ Tregs (CD8+CD25highCD127low) was also noticed among PAP vs. HC (Fig. 3f, Supplementary Figs. 2C, 3C). These findings demonstrated that type-2 CD8+ T cells charaterize the peripheral immune profile in VAP while the fraction of CD45RA+ CD8+ Tregs in VAP is inferior to that in HC at baseline and during the AIT initiation phase. In short, our data underscore an unrecognized link between several CD8+ T-cell subsets and insect-venom allergy pathogenesis and possibly the long-term immune-tolerance induced by AIT.

Although CD4+ Tregs (CD4+CD25highCD127low) have been reported to play an important role in various allergic T2 diseases[43], we observed no significant difference in CD4+ Treg percentage between VAP and HC at any of our analyzed time points (Fig. 2b). Interestingly, the percentage of CD45RO+ CD4+ Tregs (memory Tregs) was significantly although modestly lower in PAP vs. HC at three of five time points (Supplementary Figs. 2C, 4A).

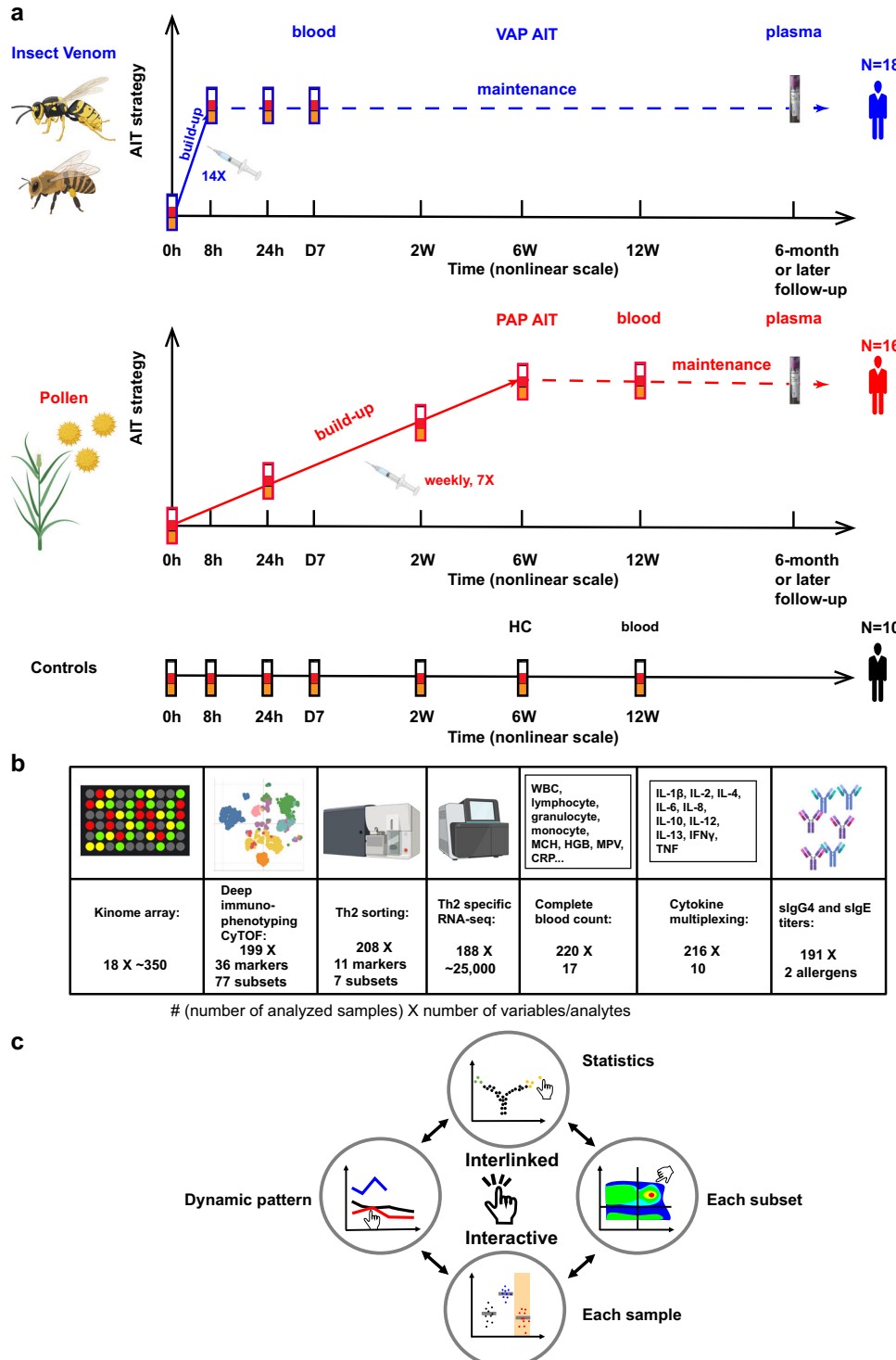

**Fig. 1 | Cohort overview, experimental schematic and interactive resource sharing. a** Overview of our longitudinal cohort SYS-T-ACT. AIT allergen-specific immunotherapy. HC, healthy controls, *n* = 10 independent individuals; PAP, pollen allergy patients, *n* = 16 independent individuals; VAP, venom allergy patients, *n* = 18 independent individuals. **b** Multilayer-Omics experimental design. 0 h, baseline; 8 h, 8 hours; 24 h, 24 hours; D7, day 7; 2 W, week 2; 6 W, week 6; 12 W, week 12 following AIT launch in patients or following the first sampling in HC. **c** Schematic of our online interactive interlinked immunological Data resource (shortened as i3Dare) to share with readers. Created in BioRender. Demczuk, A. (2024) BioRender.com/x67t048 (**a**, **b**).

Remarkably, a negative correlation between pathogenic Th2 (i.e., CRTH2+Th2 cells) and CD4+ Tregs was observed in HC, whereas a positive correlation between these two types of cells was detected in patients, especially in PAP (Supplementary Fig. 2G). This correlation shift indicates that CD4+ Tregs can successfully suppress pathogenic Th2 cells in HC, which became impossible in both patient groups, as Th2 responses escape Treg control due to different antigen specificities in respiratory allergic diseases[44]. In agreement with that scenario, in peanut-allergy patients, more evidence for a multi-functional Th2 response than a Treg insufficiency was also documented[45]. The data in our cohort show that although the fraction of CD4+ Tregs does not differ in

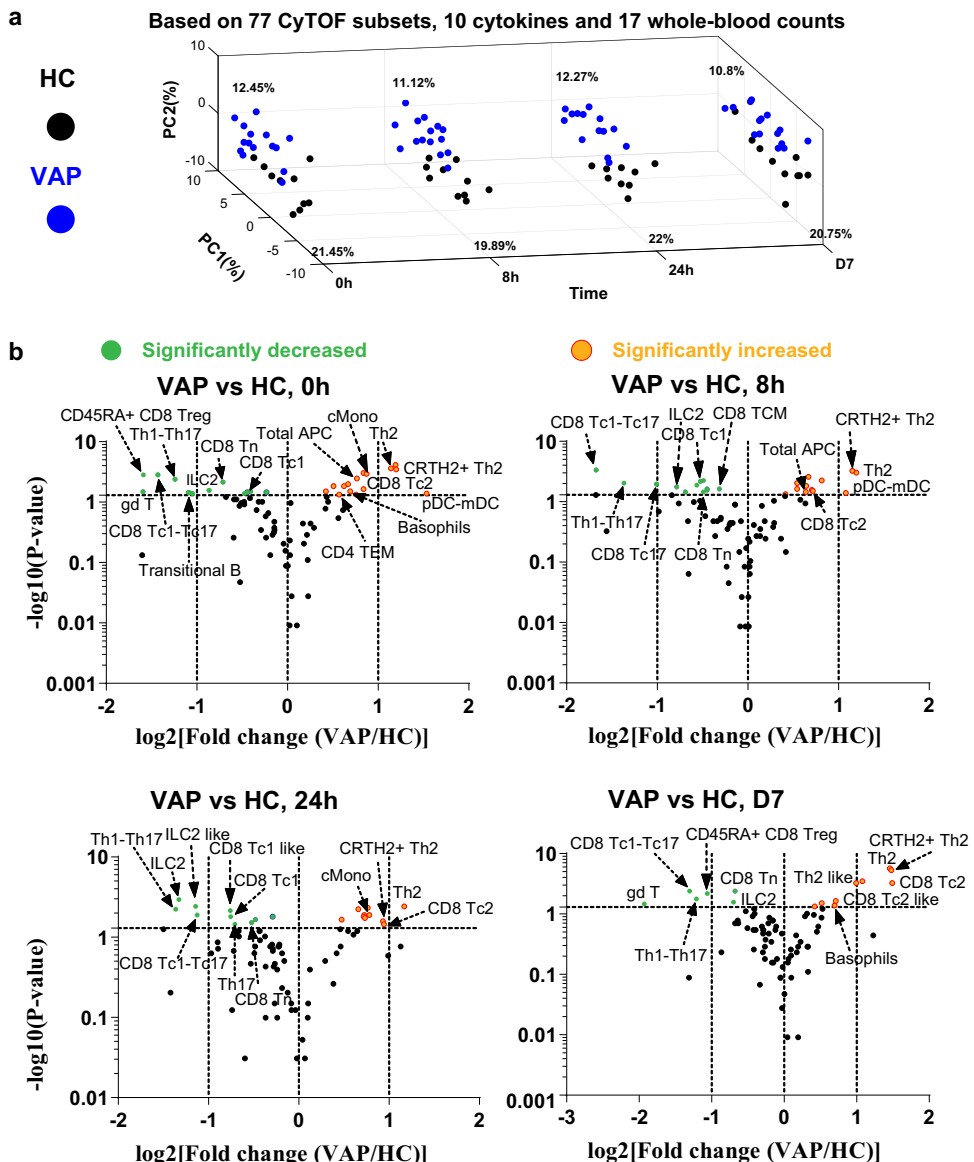

**Fig. 2 | First-week immune responses in VAP following AIT launch. a** Time–slice PCA plot of the samples from VAP and HC throughout various time points. The analysis was performed by integrating CyTOF, whole-blood-count and cytokines data. **b** Volcano plot showing the comparison of the percentages of different immune subsets, as quantified by supervised analysis of CyTOF data in VAP vs. HC at the indicated sampling time point. The selected list of significantly enhanced or decreased immune subsets (*p* < =0.05) are highlighted. Too general subsets (e.g., CD3⁻CD19⁻) were not labelled even if they showed a significant change. APC, antigen-presenting cells; CD4⁺ Th1, type 1 helper T cells; CD4⁺ Th2, type 2 helper T cells; CD8⁺ Tc1, type 1 cytotoxic cells; CD8⁺ Tc2, type 2 cytotoxic cells; CD8⁺ Treg, CD8⁺ regulatory T cells; cMono, classical monocytes; DC, dendritic cells; gdT (γδ T), gamma-delta T cells; ILC, innate lymphoid cells; mDC, myeloid DC; pDC,

plasmacytoid DC; pDC-mDC, hybrid plasmacytoid and myeloid DC; Tc1-Tc17, hybrid type 1 and type 17 cytotoxic CD8⁺ T cells; Th1-Th17, hybrid type 1 and type 17 CD4 T helper cells; TCM, central memory T cells; TEM, effector memory T cells; Tn, naive T cells. HC, healthy controls, *n* = 10 independent individuals (**a**, **b**); PAP, pollen allergy patients, *n* = 16 independent individuals (**b**); VAP, venom allergy patients, *n* = 18 independent individuals (**a**, **b**). *P*-value was determined by non-paired two-tailed Mann-Whitney test without the adjustments for multiple comparisons. The full lists of immune subsets with marker combinations, fold changes and *P* values generated from the comparisons between different groups and time points by our supervised CyTOF analysis in volcano plots of this and other figures were also provided as part of Source Data. For more details, one could visit our i3Dare website. Source data are provided as a Source Data file.

VAP vs HC before and during the AIT initiation phase, the equilibrium between CD4⁺ Tregs and Th2 is notably dysregulated in VAP.

### Multiple aberrant hybrid subsets characterize the initiation phase of AIT

Among the decreased subsets in VAP vs. HC, the CD4⁺ hybrid Th1-Th17 cells (CD45RO⁺CD45RA⁻CXCR3⁺CCR6⁺), co-expressing CXCR3/CD183 and CCR6/CD196[46], unexpectedly remained low at each time point (Figs. 2b and 3g, h). Notably, the percentage of

hybrid Th1-Th17 cells (also known as Th1/17) was also lower in PAP vs. HC at all the five matched time points (Supplementary Fig. 2C). Hybrid Th1-Th17 cells have been characterized by an enhanced and persistent anti-tumor function relative to classical Th1 or Th17 cells[47,48]. Hybrid Th1-Th17 cells are also associated with late-forming and low-burden granulomas in tuberculosis control[49]. So far, no role for hybrid Th1-Th17 cells has been attributed to the control of Type 2 immune diseases. A lower level of these hybrid cells was independently confirmed in the unsupervised analysis (Fig. 3i). Even more surprisingly, the percentage

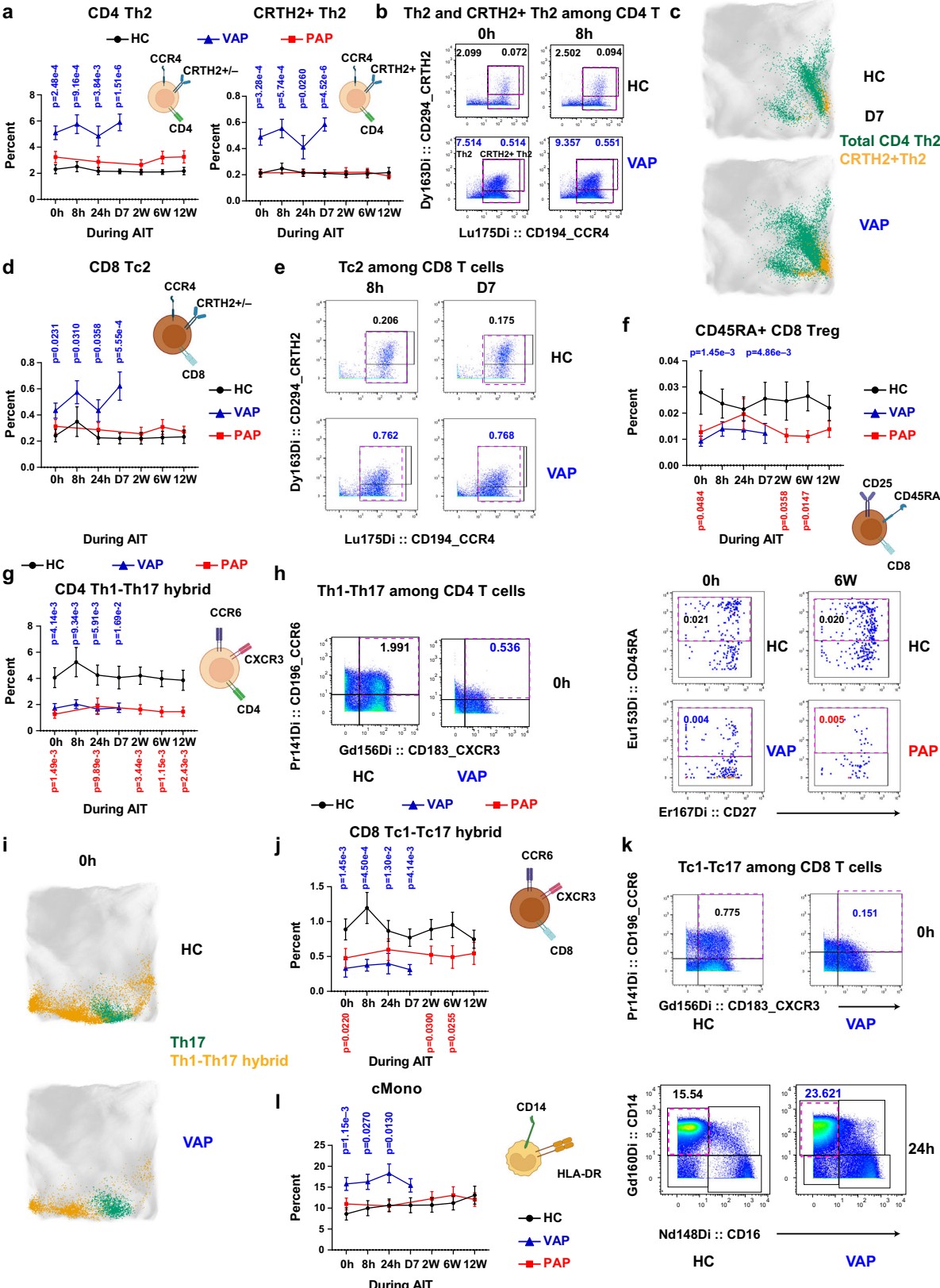

of the CD8[+] counterparts (CD8[+] hybrid Tc1-Tc17 cells, also known as Tc1/17, CD45RO[+]CD45RA[-]CXCR3[+]CCR6[+]) was lower in VAP vs. HC at all the time points (Fig. 3j, k). This also held true in PAP at three of five time points (Supplementary Fig. 2C).

As expected[50], the percentage of basophils was significantly higher in VAP vs. HC at several time points, from baseline on

(Fig. 2b, Supplementary Fig. 4B). In PAP vs. HC at several time points, basophil-like cells, defined as the direct parent gate of basophils (Supplementary Figs. 1, 2C and 4C), were significantly more plentiful together with CD8[+] Tc2-like cells. However, the percentage of CD63[+] cells among basophils (i.e., activated basophils) that were very low ex vivo did not differ between both VAP and PAP groups and HC, not

**Fig. 3 | Hybrid subsets mark immune signatures in VAP during the AIT initiation. a** Percentages of total Th2 and CRTH2[+] Th2 cells among living singlets. q = 5.00e-4 or q = 1.65e-4 (0 h), q = 1.23e-3 or q = 1.93e-4 (8 h), q = 3.88e-3 or q = 6.57e-3 (24 h), q = 6.08e-6 or q = 4.56e-6 (D7) – VAP vs HC for total Th2 and CRTH2[+] Th2 cells, respectively. Representative cytometry plots of CD194/CCR4 and CD294/CRTH2 expression among CD4[+] (**b**) or CD8[+] (**e**) non-Treg CD45RA[-]CD45RO[+]CCR6[-]CXCR3[-] cells. Percentage of total or CRTH2[+] Th2 cells as highlighted by dashed purple frames is indicated in the left or right side within the given plots. Representative EmbedSOM plots highlighting Th2 and CRTH2[+] Th2 cells (**c**) or Th17 and hybrid Th1-Th17 cells (**i**). **d** Percentages of CD8[+] Tc2 among living singlets. q = 0.0271 (0 h), q = 0.0271 (8 h), q = 0.0271 (24 h), q = 1.68e-3 (D7) – VAP vs HC. **f** Percentage of CD45RA[+] CD8[+] Tregs (CD8[+]CD25[high]CD127[low]CD45RA[+]) among living singlets. q = 2.93e-3 (0 h), q = 4.91E-3 (D7) – VAP vs HC; q = 0.0815 (0 h), q = 0.0815 (W2), q = 0.0743 (W6) – PAP vs HC. Representative plots of CD45RA and CD27 expression among CD8[+] Tregs. **g** Percentage of CD4[+] hybrid Th1-Th17 cells among living singlets. q = 0.0119 (0 h), q = 0.0126 (8 h), q = 0.0119 (24 h), q = 0.0171 (D7) – VAP vs HC; q = 3.77e-3 (0 h), q = 9.99e-3 (24 h), q = 4.34e-3 (2 W), q = 3.77e-3 (6 W), q = 4.09e-3 (12 W) – PAP vs HC. Representative plots of CD196/CCR6 and CD183/CXCR3 expression among CD4[+] non-Treg (**h**) or CD8[+] non-Treg (**k**) CD45RA[-]CD45RO[+] cells. Percentages of CD8[+] hybrid Tc1-Tc17 cells (**j**) or cMono (**l**, CD3[-]CD56[-]CD19[-]CD20[-] HLA-DR[+]CD14[+]CD16[-]) among living singlets. Right panel in **l**, Representative plots of CD14 and CD16 expression among APCs. q = 7.31e-4 (0 h), q = 4.54e-4 (8 h), q = 3.27e-3 (24 h), q = 1.39e-3 (D7) – VAP vs HC; q = 0.0504 (0 h), q = 0.0504 (W2), q = 0.0504 (W6) – PAP vs HC (**j**); q = 3.49e-3 (0 h), q = 0.0273 (8 h), q = 0.0196 (24 h) and q = 0.0458 (D7) – VAP vs HC (**l**). Data represent mean ± standard error of mean (SEM) (**a, d, f, g, j, l**). *P*-value in **a, d, f-g, j, l** was determined by non-paired two-tailed Mann-Whitney test. q-values were generated using the two-stage step-up method (Benjamini, Krieger, and Yekutieli). Unlabeled, not significant (*p* > 0.05). For all the panels, HC, healthy controls, n = 10 independent individuals; PAP, pollen allergy patients, n = 16 independent individuals; VAP, venom allergy patients, n = 18 independent individuals. Source data are provided as a Source Data file. Created in BioRender. Demczuk, A. (2024) BioRender.com/o66n384 (**a, d, f, g, j, l**).

only at baseline, but also following AIT launch (Supplementary Fig. 4D). Such a low percentage of activated basophils in patients might be explained by a much higher dose of purified antigen needed (>10 ng/ml to induce clear CD63 expression[51]) in the in vitro functional basophil activation tests than the concentrations of the specific allergen achieved by the injected total extracts in our study (estimated only ~1 ng of specific allergen in 1 ml blood without consideration of protein degradation). Among more abundant immune subsets in VAP but not PAP relative to HC were classical monocytes (cMono, CD3[-]CD56[-]CD19[-]CD20[-]HLA-DR[+]CD14[+]CD16[-]) and total HLA-DR[+] antigen-presenting cells (APC, for the gating strategy refer to Supplementary Fig. 1) (Figs. 2b, 3l, Supplementary Fig. 4E). Unexpected, a less-characterized immune subset, the so-called hybrid pDC-mDC cells, which co-express CD123 and CD11c, was also significantly higher in VAP at both baseline and 8 h following AIT (Fig. 2b, Supplementary Fig. 4F). This difference was not statistically significant in PAP (Supplementary Figs. 2C, 4F). The percentage of this under-explored hybrid DC subset is elevated in patients with acute myeloid leukemia with the duplication of the oncogene FLT3[52]. FLT3 and its ligands play critical roles in DC development[53,54]. Our data highlight an unexpected abundance of cMono and one hybrid APC subset (i.e., hybrid pDC-mDC cells) during the AIT initiation phase starting at baseline in VAP.

Another diminished subset were ILC2[55], which were lower in VAP than HC at each time point (Fig. 2b, Supplementary Fig. 4G), as well as in PAP at several time points (Supplementary Fig. 2C). In line with our ILC2 data, the percentage of IL-10[+]KLRG1[+] ILC2 is lower in patients with grass pollen allergy[12]. Our data indicate that ILC2 might play a critical role related to immune pathogenic responses not only in PAP, but also in VAP. In addition to ILC2, another bridging subset between innate and adaptive immunity[56,57], namely γδ[+] T cells, showed a lower percentage in VAP vs. HC at several time points (Fig. 2b, Supplementary Fig. 4H). Such a reduction was not statistically significant in PAP vs HC (Supplementary Figs. 2C, 4H).

### Immediate pulse of Bregs accompanied with temporary contraction of allergen-specific Th2 in VAP following AIT launch
We next analyzed which specific immune subsets were altered in VAP immediately following AIT onset vs. baseline. No significant cellular change was observed at any time points except for 8 h and 24 h following AIT launch. The only subset significantly upregulated at 8 h vs. baseline (Fig. 4a, b) was late transitional B cells as defined by the combination of CD24 and CD38[58]. Before AIT, the percentage of late transitional B cells (CD24[++]CD38[++]) was lower in VAP vs. HC (Fig. 4b). Since the regulatory protective role of transitional B cells in kidney allograft rejection has been reported[59,60], the enhanced percentage of transitional B cells in our study might contribute to rapid tolerance induction following AIT. As human regulatory B cells (Bregs) are enriched in CD24[high]CD38[high] transitional B cells[61], we further analyzed IL-10-producing Bregs in our cohort. Notably, following in vitro CpG stimulation (Fig. 4c), CD19[+]CD3[-] B cells from VAP at 8h indeed contained a significantly higher fraction of IL-10[+]TNF[-] cells than that at baseline (Fig. 4d, e), indicating an immediate boost of peripheral Bregs in VAP following AIT launch. The ratios between IL-10[+]TNF[-] and IL-10[-]TNF[+] cells, indicative of the suppressive capacity of B cells[62], were also increased at 8 h vs. baseline in VAP (Fig. 4d). As expected, no significant change in Bregs was observed in HC samples after 8 h compared to baseline (Fig. 4d). At the same time, we noticed an expression pulse of the key activation marker CD25 among memory B cells, total CD45RO[+] memory CD4[+] T cells, CD4[+] Th1 and Th2 cells immediately at 8 h vs. baseline among VAP following AIT (Fig. 4f). A significant increase in the expression of the key functional marker HLA-DR was noticed in antibody-secreting cells, i.e., plasma cells (Fig. 4f). At 8 h, a transient increase of CD127 among CD8[+] Tc1 was observed (Fig. 4f). It has been well-established that CD127 (IL-7R) plays a non-redundant role in the survival of memory CD8[+] T cells[63]. The temporary induction of CD25 expression at 8 h was followed by the wave of the key costimulatory receptors, e.g., upregulation of CD27 among memory CD4[+] T cells, CD4[+] Th1, CD4[+] Th2 cells and CD8[+] Tregs within VAP at 24 h (Fig. 4g). Importantly, the crucial antigen-presenting molecule HLA-DR was significantly but only transiently upregulated in non-classical monocytes (ncMono, CD3[-]CD56[-]CD19[-]CD20[-] HLA-DR[+]CD14[-]CD16[+]) (Supplementary Fig. 1) in VAP at 24 h following AIT launch (Fig. 4g).

The finding of a temporary activation of several key extracellular markers of B cells and CD8[+] T cells in VAP prompted us to ask whether the downstream signaling pathways are also affected during the early course of AIT. Encouragingly, following the unbiased kinome measurements of ~350 kinases in peripheral blood mononuclear cells (PBMC) depleted of CD4[+] T cells, several PKC (protein kinase C) isoforms, especially PKC-α and PKC-β, which play a critical role in regulating T and B-cell activation and survival[64], were upregulated at 8 h vs. baseline in VAP (Fig. 4h, Supplementary Fig. 5). Several cyclin-dependent kinases (CDKs), vital for cell-division regulation and actin cytoskeleton organization, also showed increased activity, indicative of cellular activation, active migration and cell-cell interaction. Several essential Ca2 + /calmodulin-dependent protein kinases, including AMP-Activate Kinase Alpha 1 Subunit (AMPKa1), a well-known anti-inflammatory gene[65] and regulator of metabolic homeostasis, were also among top-ranked activated kinases, signifying the general activation of peripheral immune cells immediately following AIT launch. Although our kinome analysis was performed without prior sorting of each immune subset that would otherwise allow to pinpoint the exact cellular origin of the observed signaling cascades, it supports the overall notion that AIT immediately boosts activation of

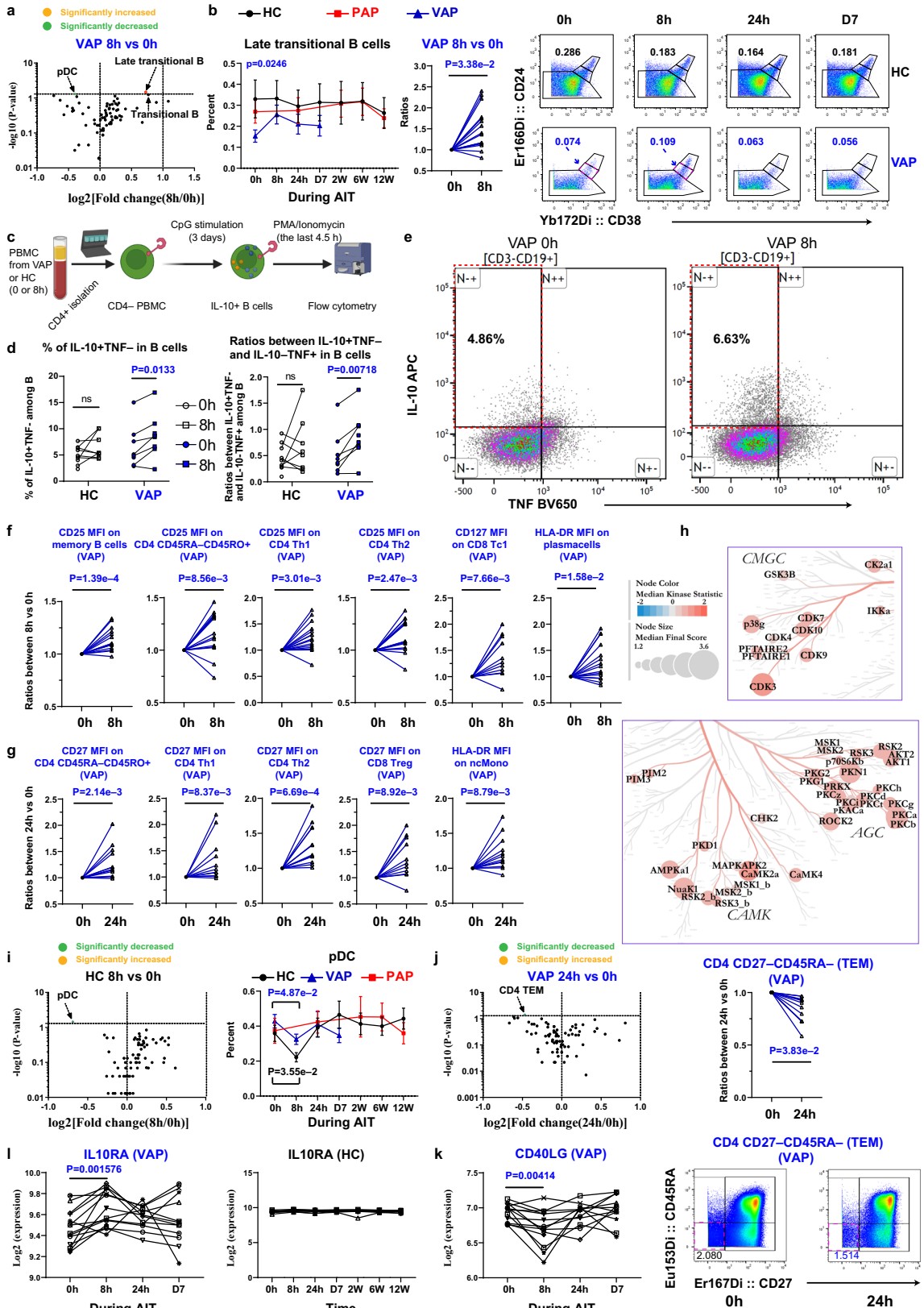

several types of peripheral immune cells in VAP during the early up-dosing phase.

The only decreased subset at 8 h vs. baseline among VAP were pDC (plasmacytoid dendritic cells, HLA-DR⁺CD11c⁻CD123⁺, Fig. 4a). Notably, such a decrease was also observed among HC in the samples taken in the afternoon (4 pm) as compared to those sampled eight

hours earlier in the morning of the same day (8am, Fig. 4i, please also refer to Methods). No change in pDC was observed at other later time points vs. baseline among HC when we sampled strictly at the identical early morning hour for each HC as at baseline (Supplementary Fig. 2H).

We also observed a temporary decrease in the percentage of CD27⁻CD45RA⁻CD4⁺ effector memory T cells (CD4⁺ TEM) at 24 h vs.

**Fig. 4 | Pulse of IL-10-producing B cells accompanied with decline of allergen-specific Th2 cells in VAP at 8 h following AIT launch. a** Volcano plot of the immune subsets percentage changes. **b** Left, Percentages of late transitional B cells (CD19⁺CD27⁻CD24⁺⁺CD38⁺⁺) among living singlets. Middle, Normalized to baseline. Right, Representative cytometry plots of CD24 and CD38 expression among CD19⁺CD27⁻ B cells. **c** Schematic of analyzing IL-10-producing B cells. **d** Percentages of IL-10⁺TNF⁻ cells or IL-10⁺TNF⁻ / IL-10⁻TNF⁺ cell ratios among living B-cell singlets following stimulation. **e** Representative cytometry plots of B-cell intracellular cytokine expression. **f, g** Normalized MSI of ex-vivo expression of indicated markers. **h** Kinome tree showing a significant change (8 h vs. baseline) in VAP's PBMC depleted of CD4⁺ T-cells. The nodes showing a Median Final Score (representing significance) >1.2 are marked with label(s). The Median Kinase Statistic represents the direction of effect. **i, j** Volcano plots of the immune subset percentage changes (left). Right panel of **i**, percentages of pDC (CD11c⁻CD123⁺) among living singlets. Upper right in **j**, Normalized percentages of CD4⁺ TEM (CD27⁻CD45RA⁻). Lower right in **j**, Representative cytometry plots of CD27 and CD45RA expression among baseline among VAP (Fig. 4j). As summarized elsewhere[66], CD27⁻ T cells CD4⁺ T cells. Expression of *CD40LG* (**k**) or *IL10RA* (**l**) transcripts in Th2 cells (*n* = 14 independent individuals). Data represent mean ± SEM (**b, i**). *P*-value was determined by paired two-tailed Mann-Whitney test in **a, b, i, j** or by paired two-tailed Student *t* test in **d, f, g, k, l**. q-values were generated using the two-stage step-up method (Benjamini, Krieger, and Yekutieli). For **h**, a mixed model statistical analysis was based on a one-tailed permutation test without multiple comparison correction (see Methods). ns, non-significant (*p* > 0.05). Each line in **b, d, f, g, j** and **k, l** links different time points of the same individual. In **a, i, j**, significantly enhanced or decreased immune subsets (*p* < =0.05) are marked. MSI, median signal intensity; ncMono, non-classical monocytes; pDC, plasmacytoid DC; CD8⁺ Treg, CD8⁺ regulatory T cells; TEM, effector memory T cells. For all the panels, HC, healthy controls, *n* = 10 independent individuals except for **d** and **h** (*n* = 9); PAP, pollen allergy patients, *n* = 16 independent individuals (no PAP in **d** and **h**); VAP, venom allergy patients, *n* = 18 independent individuals except for **d** and **h** (*n* = 9). Source data are provided as a Source Data file. Created in BioRender. Demczuk, A. (2024) BioRender.com/n34j539 (panel **c**).

---

baseline among VAP (Fig. 4j). As summarized elsewhere[66], CD27⁻ T cells are allergen-specific Th2 cells in allergic patients. Our results showed that CD4⁺ TEM were transiently declined early on at 24 h among VAP following the launch of ultra-rush AIT, which might also contribute to the well-known long-term beneficial outcomes in VAP. This data was in line with an increased CD27 MFI measurement in Th2 cells (Fig. 4g). Concordantly, mRNA levels of another antigen-specific marker, *CD40LG*, were significantly decreased in sorted Th2 cells at 8 h vs. baseline in VAP (Fig. 4k, Supplementary Fig. 6). Notably, the transcript of the receptor 1 of IL-10 (*IL10RA*) was highly expressed in sorted Th2 cells and was modestly increased at 8 h vs baseline in VAP while it was kept constant in HC over time (Fig. 4l). In contrast, we did not find any significant changes of any analyzed cell type in PAP at any later sampling time points vs. baseline (Supplementary Fig. 2I).

## Non-inflammatory transient induction of IL-6 signaling in CD4⁺ Th2 cells of VAP at 8 h following AIT launch

So far, we have observed a temporary and early increase of several functional markers in T cells, B cells and APCs. Following our unbiased CyTOF analysis, we indeed found Th2 as one of the most dysregulated immune subsets in VAP versus HC already at baseline and in the first week following AIT launch (Fig. 2b). Furthermore, as Th2 cells are well-known effectors of allergic responses that need to be modified through AIT, we decided to comprehensively analyze the underlying molecular changes in Th2 cells of allergic patients and performed a genome-scale RNA-seq analysis in sorted purified Th2 cells (Supplementary Fig. 6A-F and Supplementary Table 4). Cell-type specific RNA-seq analysis, showing a certain degree of granularity between bulk and single-cell RNA-seq methods, can reveal highly-accurate pseudo-single transcriptomic profiles for a given purified cell type, especially for a rare subpopulation, e.g., Th2 cells. Similar to our CyTOF results (Fig. 3a), the relative frequency of sorted ex vivo Th2 among living CD4⁺ T cells was indeed much higher in VAP vs. HC (Fig. 5a) and the relative frequency of sorted Th2 cells was highly correlated with the percentage of Th2 cells in the CyTOF analysis (Fig. 5b). Our RNA-seq analysis did not show any culprit-allergen bias (Supplementary Fig. 6D, E). All the marker genes specific for Th2 but not Th1, Th17 and Tregs were highly expressed among 10 randomly-selected Th2 samples (Supplementary Fig. 6F). Interestingly, our RNA-seq data also exhibited a diurnal distribution pattern of several key circadian regulatory genes in sorted Th2 cells. The expression level of the key circadian rhythm genes, such as *NR1D1* (also known as *REV-ERBα*) was uniformly increased in Th2 cells at 8 h vs. baseline among almost all VAP following AIT (*p*-value = 4.0E-4, Supplementary Fig. 7A). Among HC, another circadian regulatory gene, *PER3*, changed at 8 h vs. baseline (*p*-value = 0.0429, Supplementary Fig. 7B).

Unexpectedly, the number of differentially-expressed genes (DEGs) immediately peaked in VAP at 8 h after AIT onset, followed by a fast drop to a very-low level thereafter (Supplementary Fig. 7C). We then assessed whether those DEGs (Fig. 5c) are significantly enriched in any molecular pathways or processes at 8 h vs. baseline. Surprisingly, our unbiased genome-scale analysis showed that the IL-6 signaling pathway was ranked on top (Fig. 5d). We further asked which specific genes were affected in VAP at 8 h following AIT and found three genes, namely *SOCS3*, *S1PR1* and *BCL3*, were substantially upregulated, thus being displayed in the upper right corner of the volcano plot (Fig. 5e). Suppressor of cytokine signaling 3 (*SOCS3*) is a central regulatory component of the IL-6 signaling pathway[67]. Patients with genetic defects in IL-6 signaling[68], in particular those with homozygous mutations in IL-6 receptor subunit alpha (IL-6Rα) mainly present with primary atopy such as elevated IgE, eczema and eosinophilia[69]. Our observation showed a very-early and temporarily, but uniformly increased response of *SOCS3* transcripts among VAP but not HC (*p*-value = 2.85E-9, Fig. 5f). In line with the notion of enhanced IL-6 signaling after AIT launch in VAP, mRNA levels of *JAK1* and *STAT3* in the downstream pathway were also significantly upregulated at 8 h vs. baseline in Th2 cells of VAP following AIT start (Supplementary Fig. 7D, E).

Our unbiased analysis identified *S1PR1* (sphingosine-1-phosphate receptor 1) as another significantly-upregulated gene in sorted Th2 cells at 8 h vs. baseline (Fig. 5e, g). S1PR1 plays a critical role in regulating lymphocyte trafficking between blood and lymphoid organs. Although inhibiting S1PR1 slows egress of lymphocytes from lymph nodes and reduces the number of circulating lymphocytes, no obvious increase of Th2 cells was observed at 8 h vs. baseline in VAP (Fig. 3a). This might be due to the compensatory upregulation of the chemokine receptor *CCR7* (Supplementary Fig. 7F), which supports the back migration of lymphocytes from blood to lymphoid organs[70].

Our transcriptome analysis also pinpointed a substantial increase (~fourfold, Fig. 5h) in mRNA levels of *BCL3* (B-cell lymphoma 3), a member of the non-canonical NFKB pathway, in Th2 cells among VAP at 8 h vs. 0 h. IL-6 induces BCL3 expression in murine CD4⁺ T cells in a STAT3 (Supplementary Fig. 7E) -dependent manner[71]. Concurrently, the kinase IKKα that critically activates the non-canonical NFKB pathway[72] was also enhanced in PBMC depleted of CD4⁺ T cells at 8 h vs. baseline in VAP immediately following AIT launch (Fig. 4h). In line with the notion of several temporarily-activated immune subsets as indicated by various layers of our multi-omics analysis, a T-cell survival- and homeostasis-regulatory gene, *SLC29A1* (Equilibrative Nucleoside Transporter 3, *ENT3*)[73], was also located in the upper right segment of the volcano plot (Fig. 5e, Supplementary Fig. 7G). We further made use of our genome-scale ex vivo time-series RNA-seq dataset to investigate whether any other pathways were enhanced during the AIT process by slightly reducing the fold-change thresholds while only confining to those genes showing a continuous dynamic pattern (Supplementary Fig. 7H). Following this additional analysis, the detected 29 significantly increased genes were predominantly enriched in the

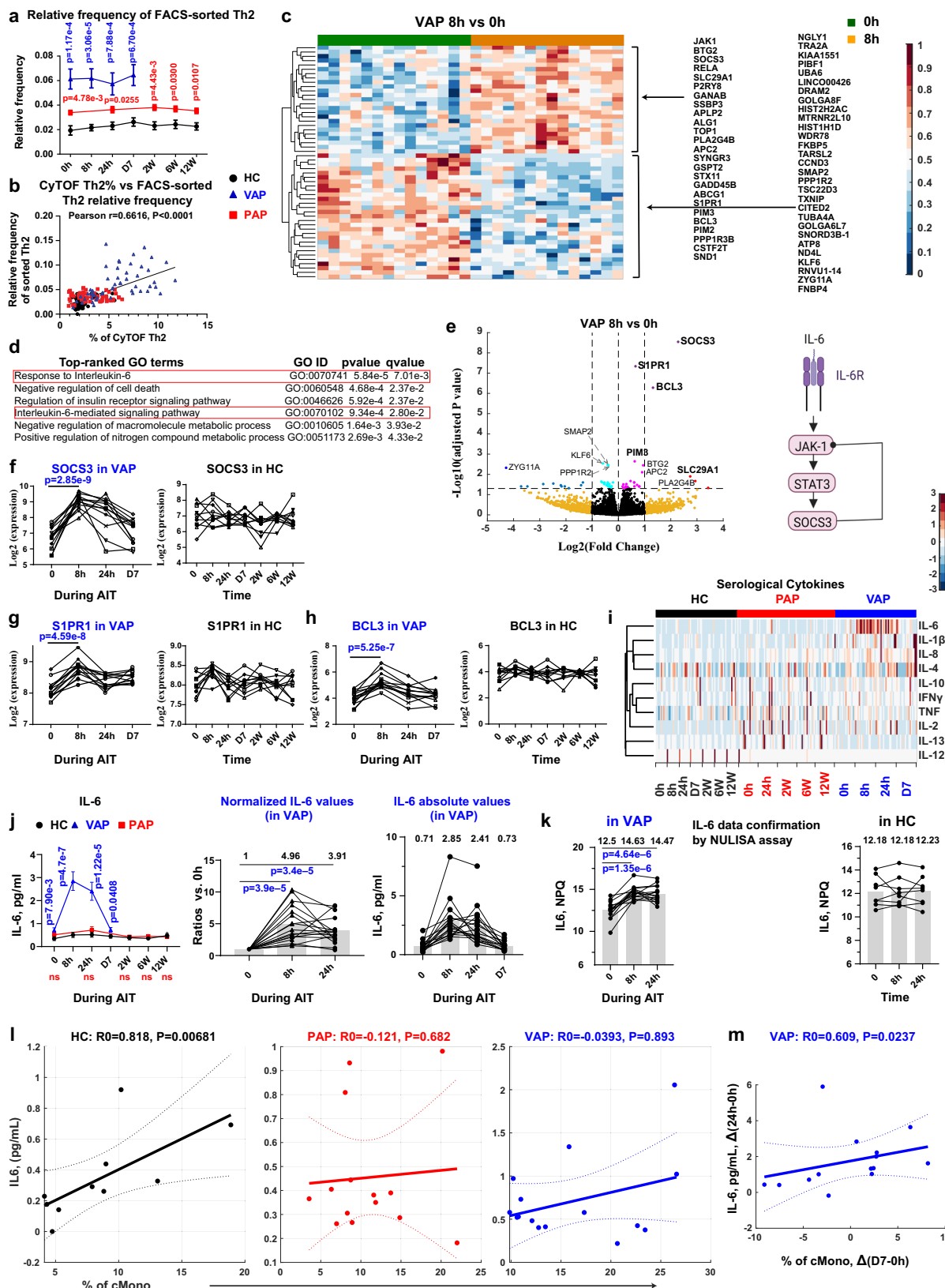

pathways regulating mitotic cell cycle, inhibiting apoptosis, regulating MAPK signaling or JAK-STAT signaling (Supplementary Fig. 7H-M). Interestingly, the unusual kinases (*PIM2* and *PIM3*), known to be induced by JAK-STAT signaling[74], were already upregulated in Th2 cells at 8 h vs. baseline (Supplementary Fig. 7J). Concordantly, as the activity of these unusual kinases is mainly regulated by expression levels but

not via post-translational modifications[74], our unbiased kinome analysis also revealed an increase in kinase activity levels of PIM2 and PIM3 (Fig. 4h) in the non-CD4[+] PBMC samples (see Methods). The other genes regulating MAPK and JAK-STAT signaling, such as *MYC*, *MAP-KAPK3*, *DUSP7* and *OSM* [another IL-6 family cytokine[75]] showed upregulated transcripts in Th2 cells one day after AIT launch

**Fig. 5 | Th2-cell-type-specific RNA-seq analysis reveals a pulse of IL-6 signaling at 8 h following AIT launch in VAP. a** Relative frequency of FACS-sorted Th2 cells among living CD4+ T cells. q = 2.37e-4 (0 h), q = 1.24e-4 (8 h), q = 7.96e-4 (D1), q = 7.96e-4 (D7) − VAP vs HC; q = 0.0121 (0 h), q = 0.0303 (D1), q = 0.0121 (W2), q = 0.0303 (W6) and q = 0.0181 (W12) − PAP vs HC. **b** Pearson correlation between the sorted Th2 frequencies and CyTOF-measured Th2 percentages. **c** Heatmap of differentially-expressed genes (DEGs) between indicated time points in Th2. **d** Top-ranked GO processes among DEGs in **c**. **e** Left, Volcano plot highlighting DEGs in Th2. The substantially and significantly changed genes were labelled. Right, Schematic of IL-6 signaling. mRNA expression of *SOCS3* (**f**), *S1PR1* (**g**) and *BCL3* (**h**) in Th2. *P*-values were adjusted with FDR. **i** Heatmap of 10 cytokines. **j** Serum IL-6 levels. Left, IL-6 levels among groups. q = 5.32e-3 (0 h), q = 9.6e-7 (8 h), q = 1.23e-5 (D1), q = 0.0206 (D7) − VAP vs HC. Normalized IL-6 levels (middle) or Scatter dot plots of absolute IL-6 levels (right). Mean for each time point in middle and right panels is labelled. **k** Confirmatory analysis of IL-6 levels by the NULISA platform. NPQ, NULISA protein quantification units (log2). Spearman correlation between serological IL-6 levels and cMono percentages among living singlets at baseline (**l**) or between the changed serological IL-6 levels and the altered percentages of cMono (**m**). Data represent mean ± SEM (**a, j**). *P*-value was determined by paired Bayes-moderated two-tailed *t* test [**c, e, f-h**] or non-paired [**a, j** (left)] two-tailed Mann-Whitney test, by paired two-tailed Student's *t* test in **b, j** (middle), **k**, by two-tailed paired Wilcoxon signed-rank enrichment test (**d**), or by two-tailed exact permutation distribution test in **l, m**. q values were generated using the two-stage step-up method (Benjamini, Krieger, and Yekutieli). NS or unlabeled, not significant (*p* > 0.05). Each line in **f-h** and **j, k** links different samples of one individual. AIT, antigen-specific immunotherapy; APC, antigen-presenting cells; CyTOF, Mass cytometry; FACS, fluorescence-activated cell sorting. For all the panels, HC, healthy controls, *n* = 10 independent individuals except for panel **k** (*n* = 8); PAP, pollen allergy patients, *n* = 16 independent individuals; VAP, venom allergy patients, *n* = 18 independent individuals except for panels **a-h, k** (*n* = 15). Source data are provided as a Source Data file. Created in BioRender. Demczuk, A. (2024) BioRender.com/t00y783 (panel **e**).

(Supplementary Fig. 7J, 7L). The genes tubulin tyrosine ligase like 12 (*TTLL12*, Supplementary Fig. 7K), regulating the mitotic phase, and immediate early response 3 (*IER3*, Supplementary Fig. 7M), reported to protect T cells from apoptosis[76], were also upregulated in Th2 cells one day later. Another MAPK signaling molecule, fibroblast growth factor 18 (*FGF18*), was increased in sorted Th2 cells one week post AIT launch (Supplementary Fig. 7M).

Having discovered the enhanced IL-6 signaling pathway in Th2 cells on the transcript level, we next asked whether serological levels of the ligand (i.e., IL-6) are also changed. Interestingly, among all the 10 analyzed Th1 (TNF, IFNγ, IL-12), Th2 (IL-4, IL-13) and other inflammation-related cytokines (Fig. 5i), IL-6 was the only one showing an uniform pattern of temporary upregulation immediately following AIT treatment among VAP (Fig. 5i, j). Although the average increase was almost five- or four-fold at 8 h or 24 h relative to baseline, respectively, the absolute IL-6 concentration (Fig. 5j) still remained in the near-homeostatic non-inflammatory range[77]. Notably, the ROC (receiver operating characteristic) analysis showed that the non-inflammatory absolute IL-6 levels alone, already at baseline (Supplementary Fig. 8A), especially at 8 h post AIT launch (Supplementary Fig. 8B), presented outstanding diagnostic power (as categorized elsewhere[78]) to distinguish VAP and HC with an area under the curve (AUC) of up to 0.9941 (Supplementary Fig. 8A-D). We further successfully validated the IL-6 data using an independent liquid biomarker assay (Fig. 5k), which combines antibody-based detection and next-generation sequencing methods[79]. Since the main producers of IL-6 are APCs and endothelial cells[80], APC-derived IL-6 plays a critical role in restricting Th2 differentiation[81] and several relevant types of APC were more abundant among VAP vs. HC (Fig. 2b), we investigated a potential correlation between serological IL-6 levels and specific circulating APC subsets or total APCs. IL-6 levels were positively correlated with the percentage of cMono only in HC but not in PAP and VAP at baseline (Fig. 5l). In line with this notion, circulating IL-6 levels were also highly correlated with the percentages of cMono, intermediate monocytes (iMono, CD3−CD56−CD19−CD20−HLA-DR+CD14dimCD16dim) and total APCs among HC, but not in VAP or PAP (Supplementary Fig. 8E-G), when samples from different time points were grouped together. Interestingly, the increased percentage of cMono at D7 in VAP relative to baseline was significantly correlated with the augmented IL-6 levels at 24 h post AIT (Fig. 5m).

In contrast to the fast response in VAP, PAP displayed a delayed increase in the number of DEGs in Th2 cells only after the completion of the up-dosing phase, i.e., at 12 W following AIT launch (Supplementary Fig. 7C), which is also consistent with the slow dynamic evolution as suggested by our CyTOF analysis in PAP (Supplementary Fig. 2D). The top-ranked genes with increased expression (*p* < =5e-4, fold change > =1.4) were significantly enriched in the pathways regulating mitochondrial functions and tRNA processing (Supplementary Fig. 7N), indicative of enhanced metabolic and translational activities, all preparing for cell activation and proliferation. The differential response dynamics on the molecular levels represented one of the key differences between VAP and PAP during the very early up-dosing phase of AIT.

## Relation to immune tolerance induction and clinical cure in VAP

As no controlled sting challenges were possible under the real-world clinical-observational protocol, we aimed to evaluate immune tolerance induction and clinical cure in VAP by other clinical and surrogate outcome measures. To assess the clinical AIT outcome in VAP, we recorded clinical reactivity in patients who have experienced field sting(s) during the AIT course or following the completion of the full 3-5-year AIT course. Notably, although a majority of the patients were not exposed to field sting(s) yet, six VAP patients reported a total number of 9 field stings with relevant insects without showing any systemic reaction, indicating a full AIT success (Supplementary Table 1). Although skin tests with insect venom extracts are not regarded as a reliable marker to distinguish cured tolerant individuals from patients with a relapse, it is generally agreed that venom skin test negativity could be used as a good surrogate marker of clinical cure in the vast majority of cases[3,82]. Thus, we also analyzed intradermal test negativity at AIT termination. Interestingly, among 17 patients, who performed skin test, only one patient, who did not report any field stings yet, showed a negative skin test. Together, these two types of clinical analyses confirmed seven cases indicative of clinical cure in our VAP cohort following AIT.

Although AIT is known to induce varying allergen-specific IgG4 (sIgG4) responses among different individual patients[83–85], enhanced sIgG4 responses are generally accepted as surrogate markers for immune tolerance induction that might also play a central protective role as blocking antibodies induced by AIT[86,87]. We analyzed sIgG4 and sIgE titers before and after AIT launch during the early build-up phase as well as at up to three long-term follow-up visits (Month 6-20). Encouragingly, almost all the 14 VAP, who had provided serum samples taken at least six months later, showed an increased sIgG4 titer in the follow-up samples, although to a varying degree with 1.5-fold up to >700-fold increases relative to the titer at baseline (Fig. 6a). During the first week of venom AIT, as expected, sIgG4 levels were kept relatively stable, although the absolute titers varied among all the individual patients (Fig. 6a). Although sIgE titers were even increased in some of VAP following AIT (Fig. 6b), the ratios between sIgG4 and sIgE were all increased between two and 720 times in the follow-up samples relative to baseline (Fig. 6c). The apparent variation in sIgG4 responses among individual VAP, although venom AIT is known to have ~95% of AIT curative rate, is consistent with the reported incapability of predicting AIT clinical outcomes in other forms of allergy only based on sIgG4 responses[88].

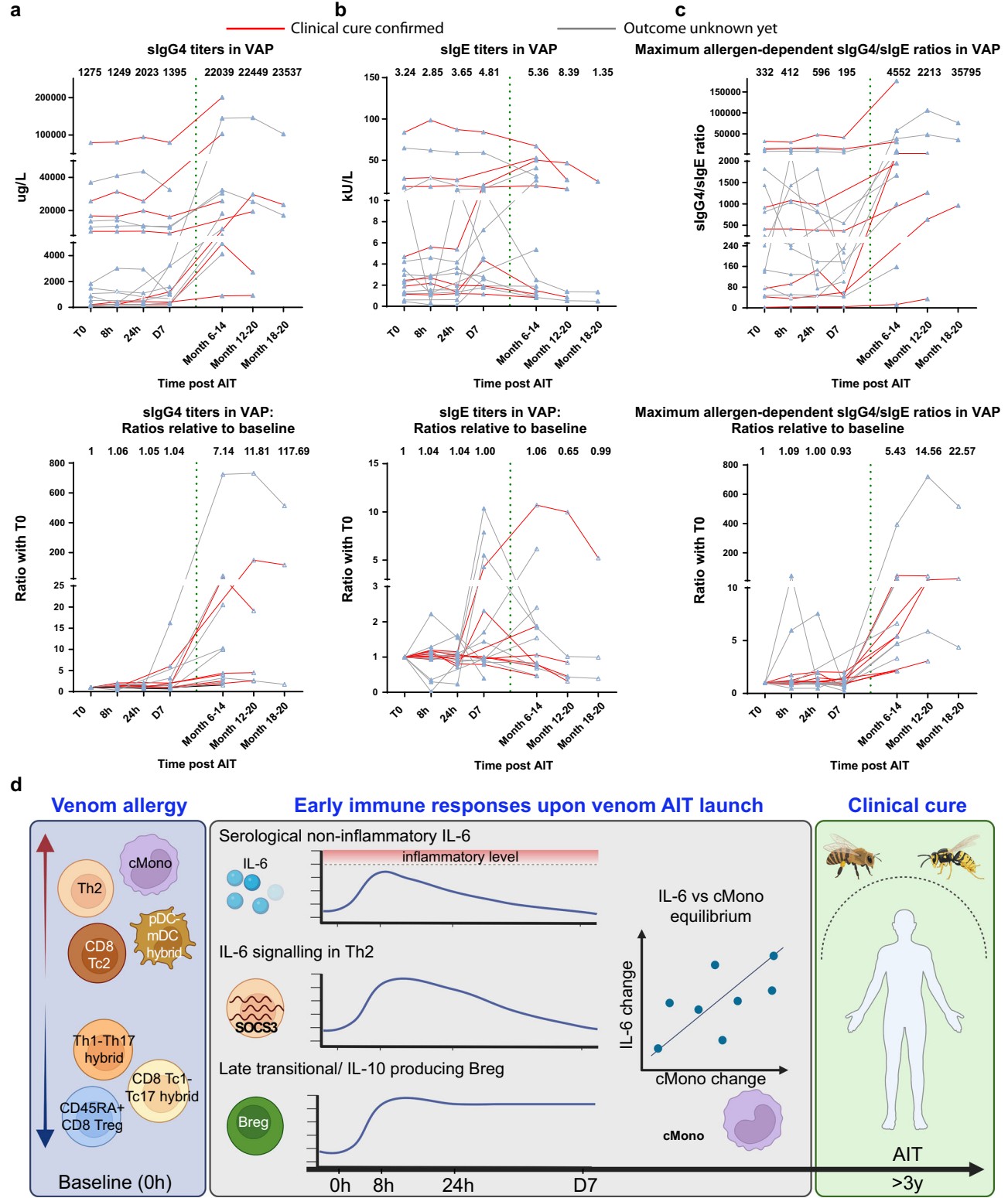

For the PAP sub-cohort, we have used a scoring method inspired by the Combined Symptom Medication Score (CSMS)[89] to retrospectively evaluate the clinical outcomes following the full 3-5-year AIT course. Different from the VAP scenarios, there were varying results. Eight out of 16 PAP (50%) were concluded unimproved while only seven (43.8%) showed an improvement and one a strong improvement (Supplementary Table 1). This heterogeneity in the AIT efficacy outcomes of PAP is in contrast to the high curative rate in VAP of our cohort. Interestingly, independent of being clinically improved or not, all PAP following birch pollen AIT or birch and grass pollen AIT showed an increase in the sIgG4 response to rBet v 1 in the follow-up samples, although with a strong variation (2-139-fold increase) (Supplementary Fig. 9). Notably, the patient with a strong improvement showed the highest absolute sIgG4 titers already at baseline. The ratios between sIgG4 and sIgE in all the PAP were also increased following birch pollen or birch and grass pollen AIT (again substantially varying between 1.8- and 5149-fold).

**Fig. 6 | AIT induces varying sIgG4 responses in VAP.** Titers of sIgG4 (**a**) and sIgE (**b**) following AIT launch in VAP. The dashed vertical line separates the long-term follow-up period from the early stage of AIT. Lower panel, the values normalized to baseline. Each triangle represents one measurement of one individual at the given time point. Each line links different time points of one individual. The red (*n* = 7 independent individuals) or grey (*n* = 11 independent individuals) line represents a clinically-confirmed case or a patient with still unknown clinical outcome, respectively. **c** Ratios between sIgG4 and sIgE titers. Lower panel, the values normalized to baseline. We here only displayed the results showing the highest ratios among the responses against two different allergens for each individual patient. The median response level for the samples of the given time point was provided. **d** Summary of our major discoveries through our systems immunology analysis in VAP following AIT launch. Red upward or blue downward arrow indicates a relatively higher or lower frequency of the indicated immune subset in VAP vs HC at baseline, respectively. VAP, venom allergy patients, *n* = 18 independent individuals (as provided in Source Data, three patients did not have long-term follow-up samples). Each line in (**a**-**c**) links different time points from one individual. AIT, allergen-specific immunotherapy; Breg, IL-10-producing regulatory B cells; CD8[+] Tc2, type 2 cytotoxic cells; CD45RA[+]CD8[+] Treg, CD45RA[+] regulatory CD8[+] T cells; cMono, classical monocytes; pDC-mDC, hybrid plasmacytoid and myeloid dendritic cells (DC); SOCS3, Suppressor of cytokine signaling 3; Tc1-Tc17, hybrid type 1 and type 17 cytotoxic CD8[+] T cells; Th1-Th17, hybrid type 1 and type 17 CD4[+] T helper cells; Th2, CD4[+] type 2 helper T cells. Source data are provided as a Source Data file. Created in BioRender. Demczuk, A. (2024) BioRender.com/l23w020 (panel **d**).

## Discussion

Insect-venom AIT results in the highest (~95%) curative rates among all AITs used for the treatment of T2 diseases. Identifying the early-stage molecular or cellular switches of insect-venom AIT could play a two-fold role: i), supporting the early decision-making whether to continue the multi-year AIT procedure or not; ii), guiding a better design of other forms of AIT in consideration of intervening with those unrecognized early molecular and cellular switches. Here we report a comprehensive analysis of the immunological response landscape during the first week of ultra-rush AIT in VAP. On the cellular level, we observed an immediate and temporary upregulation in the percentage of late transitional and regulatory B cells at 8 h vs. baseline following AIT launch in VAP that was accompanied by a transient decrease in the percentage of allergen-specific CD4[+] T cells at 24 h. At the functional level, we observed a transient upregulation of several essential activation or survival markers in the first 8 to 24 hours, such as CD25, CD27 and CD127 among several subsets of CD4[+] T cells, CD8[+] T cells and memory B cells. Notably, we also noticed a transient increase in the expression levels of the crucial antigen-presenting functional marker HLA-DR among several innate subsets (e.g., ncMono) at 24 h following AIT launch in VAP. In line with the immediate activation status in T and B cells, a kinome analysis provided further unbiased support for a general upregulation pattern of activation/survival signaling in various non-CD4[+] immune subsets among PBMC at 8 h in VAP. Meanwhile, we observed an immediate AIT-induced surge of transcripts of several key components of the IL-6 signaling pathway in sorted Th2 cells at 8 h in VAP only. The transient upsurge of cellular IL-6 signaling was accompanied by the temporary enhancement of MAPK and JAK-STAT signaling in Th2 cells as well as serological IL-6 levels. Notably, we observed a restoration of the balance between the levels of circulating IL-6 and the percentage of cMono during the initiation phase immediately following AIT launch, which was disrupted before AIT start in VAP. The strong relationship between IL-6 and the specific monocyte subset (i.e., cMono) could be partially explained by the fact that cMono secrete a much higher level of IL-6 than ncMono and iMono[90]. In contrast to VAP, we could only observe a clear transcriptional response in Th2 cells of PAP at a much later stage (i.e., only at 12 weeks following AIT launch). With our current dataset, we cannot distinguish whether the observed differential dynamic responses between VAP and PAP were attributable to different AIT allergens or due to different up-dosing schemes employed in the initiation phase, which warrants further investigation. However, the combinations of early immune responses on various layers (refer to the summary in Fig. 6d), e.g., by showing an enhanced Breg profile, non-inflammatory IL-6 signaling in Th2 cells, activation/survival signaling in various immune subsets and a rebalance between serum IL-6 levels and cMono, although being transient during the build-up time window, likely impact on the long-term beneficial outcome of AIT in VAP.

Bregs have been widely shown to play an important immunosuppressive role in various inflammatory conditions, such as autoimmune, allergic, infectious diseases and others as reported[61,91] or reviewed elsewhere[92-94]. In allergic diseases, a subset of Bregs (CD9 + B cells) have been shown to suppress both Th2 and Th17 responses in an allergic asthma animal model in an IL-10-dependent manner[95]. In our patient data, *IL10RA* was indeed highly expressed in the effector cells (i.e., Th2 cells), which was even slightly upregulated at 8 h following the ultra-rush antigen exposure in VAP. In the meantime, the antigen-specific response marker (*CD40LG*) was decreased in sorted Th2 cells at 8 h immediately following AIT launch. This indicates that Bregs might be able to suppress allergen-specific Th2 responses during AIT in VAP. It is also worth highlighting that Bregs have been shown to only transiently express IL-10 following antigen stimulation and then differentiate into antibody-secreting cells[96], which might, at least partially, explain a rapid contraction of Bregs following the intensive repeated re-exposure with venom allergens during the first 8 h period of ultra-rush AIT in VAP. Furthermore, Bregs can be reactivated upon reencounter with their cognate antigen/allergen in subsequent exposures, thereby providing long-term protection or immune tolerance during allergen re-exposures. Based on our and other Breg data in patients, we speculate that identifying and expanding allergen-specific Bregs could be applied as an allergen-specific cellular tolerance-priming therapy in other types of allergic diseases, where current AIT strategies are not sufficiently successful yet.

Our discovery that IL-6 signaling (e.g., *SOCS3*) is the major upregulated pathway in ex vivo-purified Th2 lymphocytes during the earliest time window of insect-venom AIT (8-24 h) is further supported by a uniformly detectable peak of increased serum IL-6 cytokine levels in VAP only. However, the resultant serum IL-6 levels following venom AIT launch still remained strictly within a near-homeostatic, non-inflammatory range of IL-6 in human blood (peak range in insect venom AIT at 8 h: 1.5-8.5 pg/ml; normal median healthy human plasma level: 0.75 pg/ml). Pharmacokinetic studies performed in human healthy volunteers defined a maximum physiological plasma concentration of 50 pg/ml for IL-6[77]. Although an increased IL-6 level might indicate an ongoing inflammatory response or acute infectious disease[75], IL-6 also plays an important role in regulating cellular homeostatic functions and in repressing Th2 immune responses[68,81]. In line with our data, in vitro cellular IL-6 binding studies have recently shown that only few IL-6 molecules are sufficient to activate IL-6 signaling[97]. An acute, but controlled increase in circulating IL-6 levels, as observed in our work, might transiently trigger IL-6 signaling in relevant immune subsets, such as Th2 cells, and initiate a protective tolerance response very early during AIT in VAP. In short, similar to transition states in chemistry[98], here we revealed an AIT-induced transition-state-like immune landscape (IL-6-related data and several other related observations) that is uniformly present in all VAP and might thus be a critical intermediate switch to subsequently trigger a sustainable healthy immune state. In supporting our notion that the observed very-early transition-state-like responses might have clinical implications, it is worth to note that >88% of bee venom allergic patients in whom AIT has been initiated already tolerate re-stings within one week after reaching the maintenance dose[99].

Meanwhile, our deep immunophenotyping analysis identified a general fingerprint distinguishing VAP from HC already at baseline.

This fingerprint was largely maintained in the first week during AIT. We observed more abundant total and pathogenic CD4[+] Th2 cells at baseline and during the AIT initiation phase in VAP, although recent sting events were absent. Surprisingly, our observations indicated that reduced CD8[+] but not CD4[+] Tregs, especially CD45RA[+]CD8[+] Tregs in both allergic patient groups, might also critically contribute to the pathogenesis of allergic diseases. Although different subsets of CD8[+] Tregs have been identified[100,101], their immunosuppressive functions have all been demonstrated as reported or reviewed elsewhere[102,103]. Recently, in a murine model of allergic rhinitis, adoptive transfer of CD8[+]CD25[+] Tregs has been shown to ameliorate inflammatory reactions[104]. Interestingly, a subset of CD8[+] Tregs indeed express CD45RA[105]. Therefore, a lower percentage of CD45RA[+]CD25[+] CD8[+] Tregs might play a role in promoting allergic responses in both VAP and PAP. At the same time, an unexpected strong synergy between CD4[+] and CD8[+] T cell responses was observed, i.e., both CD4[+] Th2 and type-2 CD8[+] T (CD8[+] Tc2) cells were higher only in VAP relative to controls. Notably, the percentage of both CD4[+] hybrid Th1-Th17 and CD8[+] hybrid Tc1-Tc17 cells was diminished in both patient groups, already starting at baseline. Thus, we discovered abnormal proportions of several hybrid cell types including CD4[+] Th1-Th17 and CD8[+] Tc1-Tc17 cells in both groups of allergic patients while the APC hybrid subsets, i.e., hybrid pDC-mDC cells were only upregulated in VAP. Our data firmly indicate an unexpected crucial role of several polyfunctional hybrid immune subsets with higher plasticity in insect-venom allergy. Our findings complemented another subset of plastic Th17 cells that can acquire Th2-cytokine production capacity, although reported in a murine asthma model[106,107]. Concordantly, our observations in ILC2 and γδ[+] T cells supported a particular involvement of immune cell types that play a bridging function between innate and adaptive immunity.

Moreover, from a clinical study design point of view, our work that was based on real-world patient recruitment also provided unexpected evidence for the need to consider the circadian rhythm during the sampling of immune monitoring studies, as DC have been recently shown to functionally control circadian anti-tumor responses[108]. Although another study has reported a marked decrease of pDC at 52 h (2 pm on day 3) after AIT start (8am on day 1) in patients with venom allergy[109], our results strongly suggest that the natural circadian rhythm of the peripheral distribution[23] and trafficking of immune cells is the likely cause for changes in pDC percentage of peripheral blood amongst both VAP and HC. Together, fluctuation in transcript levels of key circadian genes within Th2 cells and in percentage of circulating pDC demonstrate the necessity of considering natural circadian rhythm during sampling for clinical investigations into dynamic immune responses.

Here we analyzed ~220 blood samples over time, which represents the initial systemic effort to characterize the initiation-phase immune dynamic molecular and cellular atlas in allergic patients following AIT launch. Our multi-omics, hypothesis-free, data-driven analysis identified uniform, nevertheless yet unrecognized, cellular and molecular switches during the up-dosing phase (i.e., first week) of AIT in VAP. As this clinical AIT generates exceptionally successful long-term curative outcomes, the identified underlying early mechanisms could also be generalized to the improvements of AIT for other types of allergic diseases or T2 diseases still with lower curative rates. Moreover, optimizing the sampling time (that excludes interference by circadian rhythm) and other pre-analytical procedures in our pipeline ensures an exceptional quality of the overall datasets, which can be used as a best practice example for immunological clinical studies to reduce data heterogeneity[110,111], even in a small-scale study as demonstrated here. For ethical reasons, we were unable to recruit a placebo-controlled VAP group to benchmark our VAP AIT results. Such a placebo control group is not possible for an observational trial investigating a life-threating condition (i.e., anaphylaxis to insect stings). Thus, in our cohort, we only included healthy control participants, which represents a moderate limitation of our work. As a compensation, we have alternatively systematically analyzed the PAP group that is notably low for AIT efficacy (as defined by full clinical cure) as another negative control group. The generated rich molecular and cellular resource will provide a vigorous toolbox for gaining a systemic and dynamic comprehension of very-early immune responses during the AIT initiation phase, towards long-term curative antigen-specific immune tolerance. Mapping to our interactive reference datasets (see the i3Dare website), investigators can monitor the early immune responses of their individual patients following the AIT launch. Although T2 diseases also involve many types of non-immune cells[112], our resource will facilitate the deciphering process of T2 homeostatic immune regulation and the complex disequilibrium of immunity in various T2-associated diseases.

## Methods

### Inclusion and ethics declarations

This study complied with all ethical regulations for conducting studies with human participants and was approved by the Luxembourg National Research Ethics Committee (CNER) (SYS-T-ACT: No. 201509/11) and first pre-registered with the study protocol at ClinicalTrials.gov on 11 Oct. 2016 (NCT02931955). The experiments conformed to the principles set out in the World Medical Association (WMA) Declaration of Helsinki and the Department of Health and Human Services Belmont Report. Informed consent was obtained from each participant prior to sampling and personal data collecting, which were in full compliance with the Luxembourg National Commission for Data Protection (CNPD, before the implementation of GDPR in Europe). We obtained notification on 11 Sep. 2015 and modification approval on 17 Jul. 2017 from CNPD. No financial reward was given for VAP or PAP. Each of the ten HC received a compensation of 175€ for their time to participate the study and of 25€ for their travel costs.

We followed the global research code of conduct and included different international collaborators who meet all the authorship criteria as coauthors. We also included involved clinicians and clinical scientists in the author list.

### Cohort design, clinical diagnosis and AIT outcome evaluation

In our study, no sample size power calculation was performed in advance. At the outpatient facility of the national unit of Immunology-Allergology (Service National d'Immuno-Allergology of Centre Hospitalier de Luxembourg), which treats a great variety of allergic and immune diseases, approximately 30 to 40 new patients with anaphylactic reactions to Hymenoptera stings and around 1000 new patients with seasonal pollinosis symptoms are seen every year. After general physical examination, control of biometric data, skin and specific blood IgE testing, different treatment options are discussed with the patients. Finally, approximately half of the patients with anaphylactic reactions to wasp or bee stings will receive subcutaneous injection AIT with the relevant insect venom following an ultra-rush protocol. About half the patients with pollinosis symptoms will receive specific immunotherapy to grass- or tree-pollen. Of these the great majority of the pollen patients will routinely undergo sublingual immunotherapy and only around 10% will receive a subcutaneous pre-seasonal up-dosing protocol. In this way, a sample size of 15 per patient group was determined empirically as an achievable recruitment target at the given single clinical center over a one-year period. Eventually, the study included 18 venom allergic patients (VAP) following an ultra-rush protocol, 16 pollen-allergic patients (PAP) following a conventional up-dosing subcutaneous AIT protocol and 10 healthy controls (HC), who all consented to a complex longitudinal sampling procedure while receiving standard clinical care (VAP and PAP) or no care at all (HC). The study participants with moderate-to-severe allergic rhinitis (PAP, $n = 16$) and insect sting reactions (VAP, $n = 18$) eligible for AIT in accordance with the current international guidelines were recruited at

the national unit of Immunology-Allergology between Aug. 2016 and Feb. 2018. According to the original plan, we aimed to recruit 15 VAP with fully analyzable datasets. Due to technical issue faced during cell sorting in the first three VAP, we recruited three more VAP and eventually ended up with 18 VAP. For the first three VAP, whenever possible, different layers of analyses except for RNA-seq in sorted Th2 cells were still performed. The HC ($n = 10$) were recruited at the Clinical and Epidemiological Investigation Center of the Luxembourg Institute of Health (CIEC-LIH) between Jan. and Apr. 2018. The sample size of 10 in HC was determined empirically in consideration of the feasibility to recruit enough healthy volunteers who are willing to be sampled seven times within a 3-month period. Although no prior power calculation was made, our hypothesis-free deep immunophenotyping study of >70 different immune subsets still provided sufficient power to detect the changes of some immune subsets as several cell types indeed showed significant observations.

The study was designed as a real-world, prospective, exploratory, data-driven trial for identification of potential biomarkers and molecular switches in the early time window of AIT in the outpatient clinical setting. The inclusion criteria were the confirmed clinical diagnosis of allergy (skin prick test positivity and elevated sIgE titers to plant pollen or insect venom allergens, respectively), clinical indication for launch of AIT for the patients and absence of known allergies for healthy volunteers. The exclusion criteria were age less than 18 years, pregnancy and lactation; overt asthma or chronic obstructive pulmonary disease, concomitant autoimmune disorders, chronic diseases in exacerbation or not adequately controlled, history of hematological malignancies or solid tumors, treatment with systemic steroids, immunomodulatory agents or biologics, acute and exacerbation of chronic infections, traumas and surgeries in six months prior to the study enrollment. No restrictions were introduced regarding the number of sensitizing allergens, duration of the allergic disease, the time elapsed since the last field sting and treatment with mono- versus poly-valent allergen immunotherapy products. For VAP, we used the 4-stage Mueller grading system[113] to diagnose the anaphylaxis severity (Supplementary Table 1). For PAP, we performed the routine clinical diagnostic procedures and analyzed the comorbidity of seasonal allergic rhinoconjunctivitis (SARC), intermittent asthma (IAS), oral allergy syndrome (OAS) and eosinophilic esophagitis (EOE). We also documented the disease duration before the administration of AIT (Supplementary Table 1).

In VAP, an ultra-rush AIT protocol was administered to reach the maintenance dose within 8 hours with 14 subcutaneous injections (Fig. 1a). For PAP, a conventional up-dosing AIT was applied to reach the maintenance dose within 6 weeks with a weekly subcutaneous injection scheme. Therefore, although the exact sampling periods are different between the two groups of patients, our sampling schemes allowed us to cover comparable periods (essentially build-up phase) until the beginning of the maintenance stage for both types of AIT.

While the age of the PAP was comparable to that of HC (median, ~37 and ~35 years, respectively, Supplementary Table 1), VAP were older (median, ~49 years). In the study design, we did not consider the sex ratio as the recruitment criteria for this real-world observational study. While both patient groups included male and female participants at a comparable ratio (PAP: 8 males, 8 females; VAP: 11 males, 7 females), the HC group was predominantly female (9 of 10). Apparent gender was evaluated together with other biometric parameters as part of the routine physical examination by the clinical team before or during the recruitment process. Five of 16 PAP were sensitized to major birch pollen allergen (Bet v 1), three of 16 exhibited elevated sIgE and clinical reactivity to timothy grass pollen allergens (Phl p 1 & Phl p 5) and eight were sensitized to the pollen of both species. Most VAP (14 of 18) presented with a history of systemic reactions to wasp (yellow jacket; Vespula spp.) stings, being sensitized to Ves v 1 (one of 18), Ves v 5 (eight of 18) or both allergens (five of 18). Three of four bee VAP

showed elevated sIgE levels to both Api m 1 and Api m 10, while one VAP was mono-sensitized to Api m 1. Two VAP are hobby beekeepers. In this real-world study, no self-selection bias was introduced because no sex ratio, race and age of adults were considered during the recruitment stage.

The primary outcome of this observational study is the comprehensive immunological cellular characterization during the AIT initiation phase, whereas the secondary outcomes are the serological and Th2-cell-type-specific transcriptomic analyses. On the first day of the ultra-rush protocol, most of the VAP received their first subcutaneous injection at 8:00 AM with a gradually increasing dose in μg (i.e., starting from 0.003 to 0.006, 0.015, 0.030, 0.060, 0.150, 0.30, 0.60, 1.50, 3.00, 6.00, 15.00, 30.00, ending with 55.00) at 15-min intervals. While the baseline (0 h) blood sample was drawn at 8:00 AM, the second blood sample (8 h) was taken at 16:00. There were a few exceptions, where the first injection was given at 8:15 AM and then the afternoon sampling was delayed to 16:15. Biological samples were collected by trained study nurses, transported at room temperature (RT) within maximum two hours to the bio-specimen laboratory of the Integrated Biobank of Luxembourg (IBBL), where they were either used freshly in the sorting experiments on the same day, or cryopreserved in the IBBL biorepository for further analysis later on.

17 out of 18 VAP have completed the full 3-5-year AIT (Supplementary Table 1) except for one patient who stopped around half a year later due to the comorbidity with rheumatoid arthritis. 14 of 18 VAP have collected at least one long-term (>6 Months) follow-up plasma sample (for details, please refer to Source Data). The curative outcome was essentially evaluated[3] based on the reaction to the reported field sting(s). Although skin tests with insect venom extract are not regarded as a reliable marker to distinguish cured tolerant individuals from patients with a relapse, it is generally accepted that a negative venom skin test could be used as a good surrogate marker of clinical cure in the vast majority of cases[3,82]. Thus, we also analyzed skin test negativity at AIT termination.

Only one PAP stopped the AIT approximately two years after the onset for personal reasons while everyone else in the PAP group has finished the full 3-5-year AIT course. 14 out of 15 assayed PAP have provided at least one long-term follow-up plasma sample (for details, please refer to Source Data). The allergy specialists retrospectively evaluated the outcomes of pollen AIT for each patient after each pollen season during the 3-5 years mainly by applying the principles of the Combined Symptom Medication Score (CSMS)[89] for the peak of the last pollen season before inclusion and the last pollen season under pollen AIT (Supplementary Table 1).

## Isolation and cryopreservation of PBMCs

PBMCs (peripheral blood mononuclear cells) were obtained from fresh whole blood by density gradient immediately followed by CD4$^+$ T-cell isolation and Th2 sorting or cryopreservation. For PBMC isolation and deep immunophenotyping analysis, we followed a similar procedure as we described recently in another human cohort study[114]. To ease the comprehension, we described the major steps here again. Briefly, we collected up to 50 ml of blood per patient at each time point. 40 ml were collected in BD ACD tubes for PBMC analyses and 10 ml in BD EDTA tubes for whole-blood-count and cytokine measurements. We first added 13 ml of Ficoll Paque Plus (GE17-1440-02, Merck) at the bottom of the Falcon tubes. Blood from ACD tubes was mixed and split into two 50-ml Falcon tubes that were diluted with up to a total volume of 35 ml with DPBS (14190144, Thermo Fisher Scientific). Diluted blood was added carefully on the top of Ficoll-containing Falcon tubes. The Falcon tubes were then centrifuged at 400 g at room temperature (RT) for 30 min. The PBMC layer was collected, diluted with DPBS plus 2% heat-inactivated FBS (fetal bovine serum, 10500-064, Thermo Fisher Scientific) and centrifuged twice at 300 g for 10 min at RT. Purified PBMCs were either cryopreserved using 90% heat-inactivated FBS plus

10%DMSO (D2650, Sigma Aldrich) freezing medium and stored in liquid nitrogen or used freshly to isolate CD4+ T cells as described below. Whole-blood-count data were measured by ABX Micros CRP 200 (Horiba, Axonlab) and plasma were isolated from fresh blood collected in 10 ml BD EDTA tubes. We have also collected fecal samples over time, the analysis of which is however out of scope of the current manuscript.

## Single-cell mass cytometry (CyTOF) measurement, analysis and visualization

Peripheral blood mononuclear cells (PBMCs) were isolated and stored in liquid nitrogen until the day of the staining when the cryovials were thawed using warm DMEM 4.5 g/L Glucose with L-Glutamine (BE12-604F, Lonza) containing 5% of FBS (10500-064, Thermo Fischer Scientific) and 2 mM EDTA (15575-038, Invitrogen). Cells were then washed with PBS (17-516 F, Lonza). Prior to the metal-conjugated antibody staining and for assessing their viability, cells were re-suspended at a concentration of $1 \times 10^7$/ml and were incubated for 5 min at RT with Cell-ID Cisplatin (201064, Fluidigm, now called Standard BioTools) at a final concentration of 1 µM. The incubation was stopped by adding the staining buffer (PBS, 5%FBS, 2 mM EDTA; of note, EDTA was not used in other scenarios unless specified). Surface staining was performed by adding a cocktail of pre- or in-house-conjugated antibodies (Supplementary Table 2) for 30 min at RT; excess antibodies were removed by washing (400 $g$, RT, 10 min). It is important to mention that each tube was then split into half for introducing intense washing steps since the samples had initially shown strong signals of iodine. As a last step, samples were incubated with Ir-Intercalator (201192B, Fluidigm), diluted in MaxPar Fix&Perm (201067, Fluidigm) at a final concentration of 50 nM, and rested at 4 °C until the day of the acquisition. Prior to the acquisition, cells were washed twice with PBS and with de-ionized water two times. Centrifugation conditions after fixation, on the acquisition day, were 800 $g$, for 10 min at 4 °C. Cells were re-suspended at $5 \times 10^5$ per ml in 1:10 calibration beads (EQ Four Element Calibration Beads, 201078, Fluidigm), diluted with de-ionized water and the samples were analyzed with the Helios mass cytometer (Fluidigm) at a flow rate of 0.030 ml per min. Generated fcs files were normalized with the HELIOS acquisition software (v 7.0) by using EQ beads as a standard. Unless otherwise specified, the indicated percentage of the given subset was always relative to the total living singlets (for the full gating strategy, please refer to Supplementary Fig. 1). Statistical analysis of supervised CyTOF analysis was done in Qluocore Omics Explore v 3.8 (1). The non-paired two-tailed Mann-Whitney test was applied to compare the immune subsets between different groups (HC, VAP and PAP) at the given matched time point while paired two-tailed Mann-Whitney test was used to compare the AIT response or natural immune fluctuations between a later time point and baseline from the same patients or controls. All the precise statistical analysis methods were also described in the corresponding figure legends. No subgroup analysis was performed within each of the three groups. We also did not perform any gender-specific analysis as nine out of 10 HC were females. For the time-course measurement, q values were generated using the two-stage step-up method (Benjamini, Krieger, and Yekutieli) from GraphPad Prism 10.1.2. The volcano plot was visualized in Graphpad Prism v9.0. Of note, due to the specific Ab staining issues, the number of samples used for TCRγδ analysis ($n = 149$) was lower than that for the other immune subsets ($n = 199$) (for details regarding the TCRγδ staining issue on which sample, participant and time point, please refer to our i3Dare website). If the data point was missing due to either technical failure of the specific Ab staining in the given samples or the lack of samples, we did not perform any interpolation or extrapolation before further analysis.

Although our CyTOF data was essentially analyzed by the supervised manual gating as described above, we also independently validated our discoveries of several selected subsets of interest (only Fig. 3c, i) by our unsupervised clustering and visualization methods. The unsupervised clustering analysis was performed using the pre-gated living singlets. Pre-gating of identifying live singlets was done in FlowJo v10. Lineage markers used for clustering have been first arcsin transformed using a co-factor of 5. Unsupervised clustering has been performed using GigaSOM (https://github.com/LCSB-BioCore/GigaSOM.jl). GigaSOM is a FlowSOM-based clustering and dimension reduction method[37], adapted for huge-scale data sets using the high performance computing system in the Julia programming language (v 1.4). Of note, the analysis and visualization was performed without down-sampling. To visualize rare populations (CRTH2+ Th2 cells and hybrid Th1-Th17 cells in Fig. 3c, i), we conducted a sub-clustering on the extracted CD4+ T cells that were first manually gated using Flowjo v10. The sub-clustering analysis utilized a 20×20 SOM-grid, 100 epochs and exponential radius decay using only cell type specific set of markers [CD196 (CCR6), CD25 (IL-2 R), CD127 (IL-7Ra), CD45RA, CD45RO, CD183 (CXCR3), CD161, CD294 (CRTH2) and CD194 (CCR4)]. Cluster cell type annotation was performed using the CyCadas tool with default settings[115]. The plot was visualized using EmbedSOM v2 (https://github.com/exaexa/EmbedSOM). The recovered subsets were highlighted in a 2-dimensional embedding (Fig. 3c, i).

## Measurement of circulating cytokine levels by the MSD assay

Ten liquid immune parameters either cytokines or chemokines (IFNγ, IL-1β, IL-2, IL-4, IL-6, IL-8, IL-10, IL-12p70, IL-13 and TNF) were measured in plasma samples (isolated from EDTA tubes) from patients and healthy controls at all the time points using a multiplexing assay [The V-Plex Proinflammatory Panel 1 (hu) assay from MSD, kit catalog K15049D-1]. The samples were undiluted and measured in duplicates. The assay was performed according to the manufacturer's instructions. Data were recorded and analyzed on a MESO QuickPlex SQ 120 instrument (software version LSR_4_0_12).

## IL-6 measurement validation by the NULISAseq assay

Samples from one HC were not measured due to the lack of consent to send his or her samples out of Europe while another HC and two VAP were not analyzed due to the lack of plasma samples at one or more time points for the first three time points. NULISAseq assays were performed at Alamar Biosciences as described previously[79]. To ease the comprehension, we described the major steps here again. Briefly, plasma samples (isolated from EDTA tubes) were stored at liquid nitrogen tanks of the Integrated Biobank of Luxembourg (IBBL) and thawed on ice and centrifuged at 10,000 $g$ for 10 min. 10 µl supernatant samples were plated in 96-well plates and analyzed with Alamar's Inflammation Panel 250X assay. A Hamilton-based automation instrument was used to perform the NULISAseq workflow, starting with immunocomplex formation with DNA-barcoded capture and detection antibodies, followed by capturing and washing the immunocomplexes on paramagnetic oligo-dT beads, then releasing the immunocomplexes into a low-salt buffer, which were captured and washed on streptavidin beads. Finally, the proximal ends of the DNA strands on each immunocomplex were ligated to generate a DNA reporter molecule containing both target-specific and sample-specific barcodes. DNA reporter molecules were pooled and amplified by PCR, purified and sequenced on Illumina NextSeq 2000. Resulting sequencing data were processed using the NULISAseq algorithm (Alamar Biosciences). The sample- and target-specific barcodes were quantified, and up to two mismatching bases or one indel and one mismatch were allowed. Intraplate normalization was performed by dividing the target counts for each sample well by that well's internal control counts. Interplate normalization was then performed using

interplate control (IPC) normalization, wherein counts were divided by target-specific medians of the three IPC wells on that plate. Data were then rescaled. Following addition of +1, the data were log2 transformed to obtain NULISA Protein Quantification (NPQ) units for downstream statistical analysis.

## Allergen-specific IgE and IgG4 measurements

Allergen-specific IgE and IgG4 (sIgE and sIgG4) were measured in a clinical laboratory setting (Center for Rhinology and Allergology, Wiesbaden, Germany). The specific antigens to be analyzed were decided based on the original clinical sIgE diagnostic results. If the original clinical diagnosis indicated a positive sIgE reaction to recombinant rVes v 1 or rVes v 5, we measured the sIgG4 and sIgE titers against both antigens in all the samples of the given patient; if the original clinical diagnosis indicated a positive specific IgE reaction to rApi m 1 or rApi m 10, we also measured the titers against both allergens in VAP. For PAP, we measured sIgG4 and sIgE against recombinant rBet v 1 in all the samples of all the PAP as 13 out of 16 PAP were treated with birch pollen AIT or birch & grass pollen AIT. The plasma samples collected until week 12 were stored in the liquid nitrogen tanks at Integrated Biobank of Luxembourg while the long-term (>six months) follow-up serum samples were stored in −80 °C freezers. Those plasma or serum samples were first diluted 2X using the ImmunoCAP IgE/ECP/Tryptase Sample Diluent (10-9360-01, Thermo Fisher Scientific). Following dilution, the samples were stored in the fridge and shipped to the clinical laboratory at 4°C on the next day. The sIgE and sIgG4 measurements were performed following the manufacturer's instructions (Siemens, IMMULITE 2000 Immunoassay) with the corresponding specific recombinant allergen rVes v 5 (i209, cat. 10713576, A670L2), rApi m 1 (i208, cat. 10368595, A45L2) or rBet v1 (t215, cat. 10368597, A89L2) provided by Siemens Healthineers. For the sIgG4 results, if the resulting values were above the higher end threshold (>50,000 ug/L in 2X diluted samples), we performed reflex remeasurement using the 100X diluted samples [pre-diluted with the ImmunoCAP Specific IgA/IgG Sample Diluent (10-9361-01, Thermo Fisher Scientific)] to obtain values within the reference range.

## Analysis of intracellular cytokines in B cells following CpG stimulation

Detection of IL-10 production in human B cells followed the procedures described elsewhere[116]. To ease the comprehension, we described the major steps here again. Nine samples were selected from those collected at baseline and 8 h per participant group (either VAP or HC). The reason why only nine samples per group were selected was that we had to analyze both Bregs and kinome simultaneously to save precious clinical samples, where the number of sample spots was limited in the available chips for the kinome analysis. We selected VAP samples based on the top ranking of the plasma IL-6 level fold-changes between 8 h vs baseline while randomly selecting nine out of ten HC samples per time point. Following the thawing of bio-banked cryopreserved PBMC depleted of CD4 T cells [refer to the section 'Isolation and cryopreservation of PBMCs'], we re-suspended those cells in RPMI complete media [RPMI-1640 (102167F, Lonza) supplemented with 100 U/ml Penicillin plus 100 μg/ml Streptomycin (15140-122, Gibco), 2 mM L-Glutamine (A2916801, Gibco) and 10% heat-inactivated FBS (10500064, Gibco)]. $8.5 \times 10^5$ cells were stimulated with 1 μM of CpG (ODN2395, Invivogen) for 72 h in 200 μl of RPMI complete media with a U-bottom 96-well plate. In the last 4.5 h before staining, 50 ng/ml PMA (16561-29-8, Sigma Aldrich), 250 ng/ml Ionomycin (13909-1 ML, Sigma Aldrich) plus 1 μl of Golgiplug Protein Transport Inhibitor (555029, BD) were added together. The cells were then washed (200 g, 10 min, 4°C) and blocked with NA/LE Fc blocker (564765, BD) for 5 min at 4°C. The cells were again washed and stained with the surface marker antibody mastermix (CD3 BUV737, 741822, BD, dilution 1:200; CD19 PercP-Cy5.5, 45-0199-42, Invitrogen, dilution 1:200; LIVE/DEAD Fixable Near-

IR Dead Cell Stain, L10119, Thermo Fisher Scientific, dilution 1:500) in BD Brilliant stain buffer (563794, BD) at 4°C for 30 min. The cells were subsequently stained with selected cytokine antibodies (IL-10-APC, 506807, Biolegend, dilution 1:50; TNF-BV650, 502938, Biolegend, dilution 1:50) using the Fixation/Permeablization kit (554714, BD) following the manufacture's recommendations. The cells were finally re-suspended in the staining buffer (2% FBS in Ca2 + /Mg2+ free PBS) and acquired in a BD LSRFortessa (v8.0.1). The Breg stimulation results were analyzed in Kaluza software (v2.1, Beckman).

## Ex vivo human kinome analysis using the PamGene array

The same nine samples of the patient groups (refer to the rationale of selecting these nine samples above) processed for Breg analysis were also utilized to asses kinase activity using PamGene's (PamGene International B.V., HH's-Hertogenbosch, the Netherlands) phospho-tyrosine kinase (PTK) and serine-threonine kinase (STK) assay. Following the thawing of bio-banked PBMCs depleted of CD4 T cells [refer to the section 'Isolation and cryopreservation of PBMCs'], cells were directly (without any additional in vitro treatment) pelleted (800 g, 5 min, 4°C), before resuspension in 1 ml ice-cold PBS and pelleted again (1000 g, 10 min, 4°C). Pellets were lysed with 45 μl of M-PER (78501, Thermo Fisher Scientific) supplemented with Halt Phosphatase Inhibitor Cocktail (78428, Thermo Fisher Scientific) and Halt Protease Inhibitor Cocktail, EDTA free (78437, Thermo Fisher Scientific) for 15 min on ice. The lysate was clarified (10,000 g, 15 min, 4°C) before preparing 10-μl aliquots, each snap frozen in liquid nitrogen before storage at −80°C. One aliquot was used for protein quantification by Bradford.

Adopting a balanced design between groups, separate aliquots of the same samples were assessed on the PTK chip (86402, PamGene) and the STK chip (87102, PamGene) using the manufacturer's protocol. From each chip, a series of images describing the phosphorylation of peptides (196 for PTK and 144 for STK) corresponding to ~350 native kinases in the applied lysate was obtained in PamStation. Using the instrument manufacturer's BioNavigator software (v6.3.67.0), data was curated and normalized by chip using COMBAT before running an upstream kinase analysis. This analysis provides the Median Kinase Statistic describing the direction of effect (activation or inhibition): the change in kinase activity in the test condition (i.e., 8 h following AIT) compared to a control condition (i.e., baseline) and is formed from the median of the peptide statistics of the set of peptides that a kinase has been associated to within a defined database. The analysis also generates the Median Final Score, the sum of the significance score and the specificity score each taking a Mixed Model statistical analysis based on a one-tailed permutation test without multiple comparison correction by randomly altering sample assignment and the peptide-kinase relationship assignment, respectively. R v4.1.1 was used to compile the results and to display them using an adaptation of the CORAL tool (https://github.com/dphansti/CORAL).

## Principal component analysis (PCA) and integrative analysis

Based on the corr function in MATLAB 2020a, the correlation coefficient matrixes within or between different datasets were calculated (e.g., within different immune subsets, between immune subsets and cytokines). Since the datasets often do not follow a normal distribution, the Spearman correlation coefficient was employed for the correlation analysis described in this section. The continuous range of correlation coefficients is (-1, +1). P values following the two-tailed permutation test were also directly generated by the corr function. When the P-value is less than 0.05, we considered that the corresponding correlation coefficient was significantly different from zero.

The principal component analysis (PCA) was performed on the default setting of the pca function in MATLAB 2020a. Considering different types of datasets, different groups of participants and different sampling time points, we divided the analysis into two

categories. One type was to analyze the sample distribution between different participant groups using different combinations of datasets at a fixed sampling time point; the second type was to analyze time evolution responses within a fixed participant group. Before the PCA analysis, all the data were centered by the zscore function so that each variable had a mean value of 0 and a standard deviation of 1. In order to compare the changes of the overall immune features of each patient following immunotherapy (or each healthy control following the defined sampling period), we calculated the average distance among pairs of the adjacent two time points of each participant in the two-dimensional PCA plots. The larger the distance is, the more the overall peripheral immune system of the given subject has changed over the sampling period either caused by immunotherapy and/or natural fluctuation. Of note, each plot in the time-slice PCA figures (Fig. 2a and Supplementary Fig. 2D) is independent and different PCA plots were drawn together only for better visualization.

### Th2-cell-type-specific RNA-seq analysis

**Sorting Th2 cells and cell-type-specific mRNA sequencing.** Following the PBMC isolation from fresh blood, CD4$^+$ T cells were first enriched with human CD4 microbeads (130-045-101, Miltenyi Biotec) following the manufacture's recommendations while the remaining fractions of PBMC were immediately cryopreserved as described above. The isolated CD4$^+$ T cells were then immediately shipped at 4 °C to our sorting flow-cytometry facility. The CD4$^+$ T cells were first stained in Brilliant Stain Buffer (563794, BD) using all the Abs with specified dilution factors and configuration settings as defined in Supplementary Table 4. Samples were incubated for 30 min at 4 °C in the dark (and briefly vortexed once after 15 min). After two washes at 350 $g$, 10 min, 4 °C, the cells were re-suspended at a concentration of 10 million per ml and aseptically sorted (100 μm nozzle at 20 psi, purity mode) in a BD Aria III sorter (for the FACS gating strategy, please refer to Supplementary Fig. 6A). After sorting, cells were centrifuged at 350 $g$, 10 min, 4 °C, pelleted and lysed in 700 μl of the Qiazol lysis reagent (79306, Qiagen). The samples were transferred into 1.5 ml Eppendorf tubes and immediately frozen at −80 °C until RNA extraction. RNA was extracted using RNeasy Micro kit (74004, Qiagen) following the manufacture's recommendations. RNA-seq was performed at the EMBL Genomics Core Facility (Heidelberg, Germany) using single-read 75 bp high output by NextSeq 500 with ~ 50 million reads/sample. The NEBNext Ultra II RNA Library Prep Kit (E7770, New England Biolabs) was used and the rRNA deletion step was included. The samples of the first three VAP were not included for RNA-seq analysis due to the technical issues in sorting (also refer to Source Data). Other missing samples were mainly caused by RNA extraction failures.

**Th2-cell-specific RNA-seq pre-processing and quality control.** Quality control of the raw RNA sequencing data was conducted using the FastQC software (v0.11.7-Java-1.8.0_162) (https://www.bioinformatics.babraham.ac.uk/projects/fastqc/). Reads were mapped to the reference human genome 38 and summarized to genes using the R Bioconductor Rsubread package (v1.32.4). One sample was excluded from further analysis because CD127 Ab was not added during staining.

**Analysis of potential confounding factors.** Clinical and demographic factors that are significantly different between groups could have a confounding effect on subsequent analyses if no correction for these factors is applied. The following statistical tests were conducted to identify potential confounding factors. For categorical variables, a Fisher's exact test was performed, whereas for normally-distributed numerical variables having equal variances, one-way ANOVA was applied. In case of a normally-distributed variable with unequal variances between the groups, Welch one-way test was used. For non-normally distributed values, the Kruskal-Walls rank sum test was

employed. Homogeneity of variances was determined using Levene's test, while normality was tested using quantile-quantile (Q-Q) plots.

**Filtering, normalization, voom transformation and quality control.** Genes that consistently had zero or very low counts were first removed using the Bioconductor edgeR package (v3.24.3). Trimmed Mean of M-values (TMM) normalization was performed on the filtered data with the edgeR package. The normalized data were voom-transformed to enable differential expression analysis with the Bioconductor limma package (v3.38.3). Principal component analysis (PCA) was performed on the voom-transformed data, and a score plot was inspected where each batch was represented by a different colored symbol. In case of no batch effect, the batches do not cluster in the PCA plot. Furthermore, a principal coordinate analysis (PCoA) was conducted with the R Stats package and the score plot was inspected to check for potential outliers.

**Differential expression and enrichment analysis.** Differentially-expressed genes (DEGs) between allergic patients and controls, or between AIT-treated samples and baseline were determined by applying an empirical Bayes moderated $t$-test on the voom-transformed data, implemented in the R Bioconductor limma package (v3.38.3). Nominal $p$-values were corrected for multiple hypothesis testing by computing false discovery rates (FDR) according to the Benjamini-Hochberg procedure and using an FDR significance threshold of 0.05. Entrez gene IDs were converted to gene symbols with the R Bioconductor org.Hs.eg.db package (v3.7.0). DEGs were visualized in heatmaps and volcano plots using the R Bioconductor ComplexHeatmap (v1.20.0) and R ggplot2 (v3.2.0) packages, respectively.

Enriched differentially-regulated pathways and GO (Gene Ontology) biological processes were determined by over-representation analysis in ConsensusPathDB (release 34, https://bio.tools/consensuspathdb) using all measured genes as background list. Pathways were selected from Reactome, KEGG, Wikipathways, Biocarta and PID. Pathways and GO biological processes with Benjamini-Hochberg corrected FDR ≤ 0.05 were considered significant.

### Interactive and interlinked resource (i3Dare) sharing

To enhance the re-usability of our valuable clinical immunology resource, we employed the business intelligence tool Tableau to integrate and share various datasets in a website[117], allowing investigators to scrutinize all our results in an user-friendly, interactive and interlinked manner. In this manuscript, we could not show the dynamic response patterns of each measured immune subset following AIT. However, in our resource website, for each measured immune subset, independent of significance levels and change folds, following the click, one can visualize and explore its dynamic response pattern following the launch of AIT in different groups. Importantly, through our interactive website, each subset in each volcano plot can be related to its dynamic response pattern over time. It is also possible to compare the dynamic responses between different immune subsets, which is not realizable for any static figures. Moreover, one can also explore immune responses of each individual patient at different time points following the launch of AIT, or benchmark the measurements of their clinical samples with our reference datasets generated from HC or VAP or PAP.

### Reporting summary

Further information on research design is available in the Nature Portfolio Reporting Summary linked to this article.

## Data availability

Both raw and processed genome-scale Th2-specific RNA-seq data have been deposited in the GEO database with the accession number

GSE240503. All the raw fcs data of CyTOF analysis are available under restricted access for academic non-commercial research via https://doi.org/10.5281/zenodo.14001917[118]. The contacted corresponding author(s) will respond to the request within 10 working days and the shared link shall be expired within one month. For commercial access of our CyTOF data, parties will be directed to an appropriate contact. Our processed large-scale interactive interlinked immunological Data resource (i3Dare) of CyTOF data allows investigators to more effectively explore or reuse our immunological datasets of each immune subset from each participant at each time point (please refer to https://public.tableau.com/app/profile/lihpublicdata/viz/i3Dare_SYSTACT_Database/SYSTACTHome; ensure to activate the full-screen mode for better navigation experience). The disaggregated sex information is also available in Source Data. All other data are available in the article and its Supplementary files or from the corresponding author upon request. Source data are provided with this paper.

## Code availability

All the scripts used for the RNA-seq data analysis are deposited in GitLab and accessible via a Zenodo repository (https://zenodo.org/records/13926742) (https://doi.org/10.5281/zenodo.13926742)[119].

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

## Acknowledgements

We acknowledge all the anonymous participants in the SYS-T-ACT cohort. We appreciate all the support of research nurses especially the coordinator Jean-Yves Ferrand at CIEC of LIH and clinical nurses at Central Hospital of Luxembourg. We also thank the processing and biorepository teams at the IBBL. We acknowledge the LIH IT service unit to support and secure our resource sharing. This work was initially supported by Luxembourg Personalized Medicine Consortium (PMC, 2015 to M.O. and F.Q.H.), followed by Luxembourg National Research Fund (FNR) PRIDE programs (11012546/NEXTIMMUNE for D.P. and G.M. coordinated by M.O., 10907093/CriTiCS for C.M.C. supervised by F.Q.H. and coordinated by J. Goncalves and 14254520/i2TRON for A.D. co-coordinated by M.O.) and individual FNR AFR program (PHD-2015-1/9989160 for N.Z. coordinated by F.Q.H.) and EAACI long-term fellowship (2016, S.F.N.B.; 2022, M.L.B). F.Q.H is supported by the FNR CORE

program (C23/BM/18115323) and HPC Bridges program (18886016). F.Q.H and M.O. acknowledge funding from the Horizon Europe project COMMUTE under the grant agreement No. 101136957. Funded by the European Union. Views and opinions expressed are however those of the author(s) only and do not necessarily reflect those of the European Union or the European Health and Digital Executive Agency (HADEA). Neither the European Union nor the granting authority can be held responsible for them. P.W. has received funding from the European Research Council (ERC) under the European Union's Horizon 2020 research and innovation program (grant agreement No. 863664) and the FNR CORE program under the grant CORE/15/BM/10404093. D.B. is supported by the FNR CORE program (C18/BM/12691266 and C21/BM/15796788). The work was further supported by the Luxembourg Government through the CoVaLux programme (16954531, M.O. and P.W.). A.P. is supported by the Action Lions Vaincre le Cancer. Some icons in our Figures were created with BioRender.com.

## Author contributions

D.P., S.F.N.B., F.C.M. and C.L.W. collected clinical information and samples. D.P., S.F.N.B., F.Hedin, A.B., A.D., C.M.C, N.Z., A.P., W.A., M.K., J.R.G., S.C., L.N.C., V.B., S.D., I.W., G.M., D.R. and O.D. contributed to experimental procedures. C.G. and L.G. designed and supervised sorting. F. Hedin performed supervised CyTOF analysis. O.D., M.L.B and I.C. contributed to sIgE and sIgG4 measurement and data analysis. E.G. performed RNA-Seq analysis. A.D. and O.H. performed unsupervised CyTOF analysis. G.M. and O.D. performed Breg analysis. J.L. performed kinome analysis. X.H. and J.R.G. performed integrative computational analysis. P.A. contributed to large-scale data management. F. Hedin and A.C. contributed to interactive resource sharing. R.S., J.Z., M.H.S., J. Gutermuth, D.B., L.K., F.B., P.W., A.C. and J. Goncalves provided supervision into the various experiments of the project. M.M. and F. Hentges coordinated the clinical work. F.Q.H. and M.O. designed and oversaw the entire project. F.Q.H and M.O. wrote and revised the manuscript.

## Competing interests

Pending patent application on the protection of predictive biomarkers for AIT efficacy (patent applicant: Luxembourg Institute of Health; inventors: F.Q.H. and M.O.; EP Patent Application No. 23192753.4 entitled "EARLY RESPONSE BIOMARKERS FOR ALLERGEN IMMUNOTHERAPY"). The remaining authors of this work declare no competing interests.

## Additional information

Dimitrii Pogorelov[1,2,18,21], Sebastian Felix Nepomuk Bode ®[1,3,4,21], Xin He ®[5,21], Javier Ramiro-Garcia ®[5], Fanny Hedin[6], Wim Ammerlaan[7], Maria Konstantinou[6], Christophe M. Capelle ®[1,2,20], Ni Zeng ®[1,2], Aurélie Poli ®[1,19], Olivia Domingues[1], Guillem Montamat ®[1], Oliver Hunewald ®[1], Séverine Ciré ®[1], Alexandre Baron[1], Joseph Longworth ®[1,5], Agnieszka Demczuk ®[1,2], Murilo Luiz Bazon[1], Ingrid Casper[8], Ludger Klimek[8], Lorie Neuberger-Castillo[7], Dominique Revets[6], Lea Guyonnet[1,9], Sylvie Delhalle ®[1], Jacques Zimmer ®[1], Vladimir Benes ®[10], Françoise Codreanu-Morel[11], Christiane Lehners-Weber[11], Ilse Weets[12], Pinar Alper ®[5], Dirk Brenner ®[1,5,13], Jan Gutermuth[14], Coralie Guerin ®[1,9], Martine Morisset ®[11,15], François Hentges[11], Reinhard Schneider ®[5], Mohamed H. Shamji[16], Fay Betsou[7,17], Paul Wilmes ®[2,5], Enrico Glaab ®[5], Antonio Cosma ®[6], Jorge Goncalves ®[5], Feng Q. Hefeng ®[1,22] ✉ & Markus Ollert ®[1,13,22] ✉

[1]Department of Infection and Immunity, Luxembourg Institute of Health (LIH), Esch-sur-Alzette, Luxembourg. [2]Department of Life Sciences and Medicine, Faculty of Science, Technology and Medicine, University of Luxembourg, Esch-sur-Alzette, Luxembourg. [3]Department of Pediatrics and Adolescent Medicine, Ulm University Medical Center, Ulm, Germany. [4]Department of General Pediatrics, Adolescent Medicine and Neonatology, Medical Centre–University of Freiburg, Faculty of Medicine, Freiburg, Germany. [5]Luxembourg Centre for Systems Biomedicine, University of Luxembourg, Belvaux, Luxembourg. [6]National Cytometry Platform, Luxembourg Institute of Health, Esch-sur-Alzette, Luxembourg. [7]Integrated BioBank of Luxembourg, Luxembourg Institute of Health, Dudelange, Luxembourg. [8]Center for Rhinology and Allergology, Wiesbaden, Germany. [9]Cytometry Platform, Institut Curie; Innovative Therapies in Haemostasis, INSERM, Université de Paris, Paris, France. [10]Genomics Core Facility, European Molecular Biology Laboratory, Heidelberg, Germany. [11]National Unit of Immunology-Allergology, Centre Hospitalier de Luxembourg, Luxembourg, Luxembourg. [12]Department of Clinical Biology/ Research Group Experimental Pharmacology, Vrije Universiteit Brussel, Universitair Ziekenhuis Brussel, Brussels, Belgium. [13]Department of Dermatology and Allergy Center, Odense Research Center for Anaphylaxis, University of Southern Denmark, Odense, Denmark. [14]Department of Dermatology, Vrije Universiteit Brussel, Universitair Ziekenhuis Brussel, Brussels, Belgium. [15]Allergy Unit, Angers University Hospital, Angers, France. [16]Immunomodulation and Tolerance Group, Allergy and

Clinical Immunology, Department of National Heart and Lung Institute, Imperial College London, London, UK. [17]CRBIP, Institut Pasteur, Université Paris Cité, Paris, France. [18]Present address: Center of Allergy & Environment, Technical University of Munich, Munich, Germany. [19]Present address: Neuro-Immunology Group, Department of Cancer Research, Luxembourg Institute of Health, Luxembourg, Luxembourg. [20]Present address: Institute of Microbiology, ETH Zurich, Zurich, Switzerland. [21]These authors contributed equally: Dimitrii Pogorelov, Sebastian Felix Nepomuk Bode, Xin He. [22]These authors jointly supervised this work: Feng Q. Hefeng, Markus Ollert. ✉e-mail: Feng.Hefeng@lih.lu; Markus.Ollert@lih.lu

