## [Transparent Peer Review file · Nature Communications]

Multomics approaches disclose very-early molecular and cellular switches during insect-venom allergen-specific immunotherapy: an observational study

Corresponding Author: Dr Feng Hefeng

Version 0:

Reviewer comments:

Reviewer #1

(Remarks to the Author)

This research study was designed to understand the early immunologic changes that occur during allergen-specific immunotherapy using venom extract as a model. Blood samples were collected at 8 and 24 hours, along with follow-up out to ~6 months to gain insights into immune alterations. Most impressively, the immune profiling was done with cutting-edge omics, including CyTOF, Th2 sorting and RNA-seq, CBC, cytokine multiplexing, and kinome arrays. Over 200 million peripheral cells were characterized.

In terms of significance to the field of allergy and immunotherapy, this study sheds light on the immunologic changes happening within hours of initiation - some expected and others not expected. It may be a blueprint to similar studies conducted with other forms of allergen immunotherapy, such as oral, sublingual, or epicutaneous immunotherapies for food allergies. Outside of the field of allergy, this study demonstrates the wealth of knowledge that can be gained by multi-omics studies and how to bring the data together to uncover novel findings.

I have a few comments for the authors to address:

1. In terms of study design and outcomes, can the authors provide allergen-specific IgE and IgG4 data? Since we know AIT modulates these parameters, it would be of interest to see how these are changed in these particular subjects during AIT. Likewise, were any skin prick tests or sting challenges conducted to ensure these subjects responded clinically to the AIT?
2. Why were the pollen and venom AIT schedules different? It seems that the cellular responses would be very different since venom AIT was done with ultra-rush and pollen was done with up dosing over six weeks. The authors should comment and note this limitation of the study more thoroughly.
3. No placebo-treated subjects were included. Rather, 10 healthy controls were used for the study as a "benchmark" for natural immune fluctuations. I think this is a moderate limitation of the study since healthy controls may have distinct immune function that differs from venom allergic patients. Please discuss this limitation and the rationale.
4. Lines 71-72, it is unclear what the authors are referring to when they say that the process of AIT mimics the frequent exposures to microbial factors in pre-school children on farms. Please edit this part or make it more clear what is meant - AIT is a specific allergen extract, not related to anything microbial.
5. Lines 158-160, can the authors provide details on the methodology for identifying pathogenic allergen-specific Th2 cells. Specifically, were the cells stimulated with the venom antigen in culture? Different venoms were used clinically, so were the same venoms used in a patient-specific fashion for cellular studies?
6. The identification of IL-10 producing Bregs at 8 hours is highly important. Although, it appears this cell population contracts rapidly. The authors should comment more on this finding in the discussion. For example, why do these cells

appear to be so short lived? Can this very early induction of Bregs really be responsible for long-term tolerance? While it would be speculative, I wonder if allergen-specific Bregs could be harnessed in a therapeutic application in venom and other types of allergies. This may be worth a comment if the authors feel inclined.

Reviewer #2

(Remarks to the Author)

The authors report the early immune changes during the initiation phase of insect-venom-specific immunotherapy vis-a-vis pollen allergen-specific immunotherapy using mass cytometry, bulk RNA-seq of sorted Th2 cells, MSD-based cytokine measurements, Kinome analysis, and clinical lab enumeration of immune cells in whole blood. Although the authors highlight the discovery of early-stage molecular and cellular switches in respective patient cohorts using multiomics approaches there are several shortcomings to this manuscript in its current format.

- 1) The rationale for the study is based on distinct durability of venom vs pollen-specific immunotherapy. It would thus be ideal to mention whether all the patients in VAP group showed long-term tolerance as opposed to the lack of it in all/majority of those in PAP group. Table S1 should have details on clinical diagnostic matrix for VAP.
 - 2) There are no starting hypotheses stated, and the manuscript appears as a loose, lengthy summary of all the comparisons made among various immune cell subsets rather than a focused, fluent, impactful report underlining clinically relevant/applicable findings.
 - 3) Allergen-specific immune responses are not investigated. Also, none of the observations have been validated via functional assays.
 - 4) At certain instances, non-significant findings/trends are mentioned alongside the significant ones in the main text. Such non-significant indicators could be a part of supplemental data.
 - 5) The P values should be adjusted for multiple comparisons and the resulting q values (if satisfy the significance threshold) should be mentioned in brackets in the context of respective main text.
 - 6) Most sections under Results include a writeup, which is more in line with the Discussion. Also, at multiple instances, there is a reference to figures from preceding result sections which mars the flow.
 - 7) Please rephrase "whole-genome RNA-sequencing" (Pg 5, line 132), "treatment-resulting peripheral immune responses" (Pg 5, line 144), "volcano-plot analysis", "outstanding diagnostic power" (Page 14, line 423), "clinical cure rates" (Page 15, Line 451).
 - 8) Notably, although CD8+ T cells....(Page 6, line 173). Please refer to PMID: 36333296, PMID: 32835695, and re-phrase.
- Minor:
- 1) Citation style not in line with Nature Communications

Reviewer #4

(Remarks to the Author)

The manuscript by Pogorelov et al., titled "Multiomics approaches disclose very-early molecular and cellular switches during insect-venom allergen-specific immunotherapy," is an interesting read, providing novel insights into immune cell subtypes involved in venom or pollen allergen-induced allergies, as well as the mechanisms potentially driving the beneficial effects of AIT.

One immediate question is whether the two different time regimens for up dosing used in VAP vs PAP are comparable. The authors rightly address and acknowledge this issue.

The non-inflammatory IL-6 induction as a mechanism by which AIT may drive the switch from Th2/Tc2/insufficient CD8+ Treg towards Breg and the restoration of IL-6/APC equilibrium is truly intriguing. Including a schematic diagram depicting this process in the manuscript would be beneficial.

While I have no other major concerns, there are some minor points that could be addressed to improve the paper.

Minor comments:

Line 87. The way this sentence reads now, suggests IgE levels are reduced while activation of mast cells and basophils is increased. Thus, please rephrase into e.g.,

This is demonstrated by a reduction in IgE levels (Michils et al. 2000, Galli et al. 2008), a decrease in the activation and degranulation of mast cells and basophils (Maintz et al. 2009, Akdis and Akdis 2014), and lower frequencies of allergen-specific Th2 cells (Wambre et al. 2017, Rudulier et al. 2019) and type 2 innate lymphoid cells (ILCs) (Golebski et al. 2021).

Fig 2A The UMAP visualization looks very blurry – could benefit from optimizing colours, contrast and resolution. Also the figure legend (annotation of cell subtypes within clusters) could be better designed in terms of size, font and colour.

Line 156. The frequency of Th2 is not increased, but at a higher level in VAP across all timepoints compared with HC

Fig 3C The visualization is again blurred. Please improve.

Line 211. The Th1-Th17 subset was not declined but at a lower level in PAP vs HC

Fig5D please correct typo in "reponse to interleukin 6"

Version 1:

Reviewer comments:

Reviewer #1

(Remarks to the Author)

The authors have improved the manuscript, particularly demonstrating the clinical outcomes (absence of allergic reactions after stings and pollen reactivity scoring) and serologic measurements of IgE, IgG4, and IgG4/IgE Ratios. The additional discussion of Bregs, as well as some of the limitations of this study are appreciated. I do not have any further comments.

Reviewer #2

(Remarks to the Author)

No further comments.

Reviewer #4

(Remarks to the Author)

The manuscript has much improved following the revision and is now suitable for publication in NCOMMs.

REVIEWER COMMENTS

Reviewer #1 (Remarks to the Author):

This research study was designed to understand the early immunologic changes that occur during allergen-specific immunotherapy using venom extract as a model. Blood samples were collected at 8 and 24 hours, along with follow-up out to ~6 months to gain insights into immune alterations. Most impressively, the immune profiling was done with cutting-edge omics, including CyTOF, Th2 sorting and RNA-seq, CBC, cytokine multiplexing, and kinome arrays. Over 200 million peripheral cells were characterized.

In terms of significance to the field of allergy and immunotherapy, this study sheds light on the immunologic changes happening within hours of initiation - some expected and others not expected. It may be a blueprint to similar studies conducted with other forms of allergen immunotherapy, such as oral, sublingual, or epicutaneous immunotherapies for food allergies. Outside of the field of allergy, this study demonstrates the wealth of knowledge that can be gained by multi-omics studies and how to bring the data together to uncover novel findings.

Reply: Thanks for your general appreciation not only on the novelty of our work, but also on the general interests of our work to other forms of allergies and even other diseases beyond the allergic fields.

I have a few comments for the authors to address:

1. In terms of study design and outcomes, can the authors provide allergen-specific IgE and IgG4 data? Since we know AIT modulates these parameters, it would be of interest to see how these are changed in these particular subjects during AIT. Likewise, were any skin prick tests or sting challenges conducted to ensure these subjects responded clinically to the AIT?

Reply: We have now systematically measured allergen-specific IgE and IgG4 (i.e., relevant sIgE and sIgG4) levels in all the available time-course samples (with 438 sIgE and 378 sIgG4 measurements in ~200 samples) including the long-term follow-up samples (as shown in the new **Figure 6**). We also now added a section to describe the related results in **line 425-468, page 14-15**. Essentially, our results show that AIT induces sIgG4 responses in all of the tested follow-up samples (6-20 Month) in venom patients, although with a strong variation among individual patients. The ratios between sIgG4 and sIgE were also increased in the follow-up samples in general, although with a high variability.

We have also added the AIT clinical outcome data (**Supplementary Table 1**) that were either based on field sting reactions or skin test negativity for VAP (venom allergic patients). No sting challenges were possible within our study protocol that was based on a real-world healthcare setting that does not foresee sting challenges as a routine procedure. None of the six patients, who experienced one or more field stings, showed any systemic reaction to the stings, indicating a 100% AIT curative rate in those patients.

Another VAP, who did not experience or report field sting(s) yet, has shown a negative skin test after venom AIT, also indicative of cure. We have now also integrated these clinical outcome data into the VAP sIgG4 and sIgE data (**new Figure 6**) using red lines to refer to those with confirmed clinical outcomes, while using gray lines to refer to those so far without a confirmed clinical outcome at the conclusion of the follow-up.

For the PAP patients (pollen allergic patients), our clinical team has applied a questionnaire inspired by the Combined Symptom Medication Score (CSMS) to retrospectively evaluate the AIT outcomes of the baseline and all subsequent pollen seasons under AIT (refer to cohort design of **Methods**). Different from the VAP scenario, there are varying results. Eight out of 16 PAP (50%) were concluded “unimproved” while only seven showed an improvement and only one with a strong improvement. This high heterogeneity in the AIT outcomes of PAP is sharply contrasted to the 100% curative rate in the VAP with confirmed clinical outcomes. We now also added these data in **Supplementary Table 1** and integrated the AIT outcome information into the sIgG4 and sIgE data with different color codes, representing different outcome categories. In line with the literature, our data also showed that the sIgG4 induction is not sufficient by itself to predict clinical improvement (**Supplementary Figure 9**).

2. Why were the pollen and venom AIT schedules different? It seems that the cellular responses would be very different since venom AIT was done with ultra-rush and pollen was done with up dosing over six weeks. The authors should comment and note this limitation of the study more thoroughly.

Reply: Thanks for pointing out this specific limitation. Although the exact AIT time schedules were different between pollen and venom patients, we aimed to cover the initial periods until the maintenance dose. The ultra-rush AIT was approved and is available for VAP, while no comparable pollen-AIT that is licensed for ultra-rush subcutaneous injection was at the time of the study design, nor is it currently available for PAP. We have now further explained this difference directly in the beginning of Results (**line 114-118, page 4**) in addition to the already existing sentences in Discussion (**line 497-500, page 16**) and Methods (**line 641-646, page 21**).

3. No placebo-treated subjects were included. Rather, 10 healthy controls were used for the study as a "benchmark" for natural immune fluctuations. I think this is a moderate limitation of the study since healthy controls may have distinct immune function that differs from venom allergic patients. Please discuss this limitation and the rationale.

Reply: Thanks for pointing out this limitation. As anaphylaxis is a life-threatening condition, ethically it is not feasible for us to include a placebo-treated control patient group to benchmark with the AIT

response. We now discussed this particular limitation related to the lack of placebo-treated controls more thoroughly in Discussion (line 591-597, page 19).

4. Lines 71-72, it is unclear what the authors are referring to when they say that the process of AIT mimics the frequent exposures to microbial factors in pre-school children on farms. Please edit this part or make it more clear what is meant - AIT is a specific allergen extract, not related to anything microbial.

Reply: Thanks for identifying this vague aspect. We now decided to delete that sentence, as it can add nothing essential to the manuscript but potentially misleading the readers.

5. Lines 158-160, can the authors provide details on the methodology for identifying pathogenic allergen-specific Th2 cells. Specifically, were the cells stimulated with the venom antigen in culture? Different venoms were used clinically, so were the same venoms used in a patient-specific fashion for cellular studies?

Reply: As described by Wambre et al (<https://www.science.org/doi/10.1126/scitranslmed.aam9171>), we used CRTH2 to mark pathogenic Th2 cells, in addition to those used to identify the general Th2 cells. We now directly added the combination of all the used markers (CD3⁺CD56⁻TCRgd⁻CD4⁺CD8⁻CD45RO⁺CD45RA⁻CXCR3⁻CCR6⁻CCR4⁺CRTH2⁺) in the first place, where we mentioned pathogenic Th2 cells. Furthermore, the full gating strategy has already been provided in **Supplementary Figure 1**. In our work, to make our dataset more physiologically relevant, we did not use *in-vitro* re-stimulation to identify pathogenic Th2 cells.

6. The identification of IL-10 producing Bregs at 8 hours is highly important. Although, it appears this cell population contracts rapidly. The authors should comment more on this finding in the discussion. For example, why do these cells appear to be so short lived? Can this very early induction of Bregs really be responsible for long-term tolerance? While it would be speculative, I wonder if allergen-specific Bregs could be harnessed in a therapeutic application in venom and other types of allergies. This may be worth a comment if the authors feel inclined.

Reply: We thank the reviewer for highlighting the importance of our Breg findings. Bregs are known to transiently upregulate the production of regulatory cytokines, such as IL-10, upon activation (<https://www.jci.org/articles/view/85113>). In our study, this is most likely due to contact with their cognate allergen(s) in VAP at 8 hours found in the injected venom extracts. Furthermore, they can be reactivated upon reencounter with their cognate antigen/allergen in subsequent exposures, thereby providing long-term protection or tolerance against allergen re-exposure. One could speculate that

identifying and expanding allergen-specific Bregs could be used as an allergen-specific cellular tolerance therapy.

We have now added one paragraph in Discussion (line 506-523, page 16-17, also see that paragraph below here) to particularly discuss about the Breg related data and literatures. Of note, to better interpret our data, we have now also displayed the IL-10RA expression data on Th2 cells (Figure 4L).

Bregs have been widely shown to play an important immunosuppressive role in various inflammatory conditions, such as autoimmune, allergic, infectious diseases and others as reported^{61, 92} or reviewed elsewhere^{93, 94, 95}. In allergic diseases, a subset of Bregs (CD9+ B cells) have been shown to suppress both Th2 and Th17 responses in an allergic asthma animal model in an IL-10-dependent manner⁹⁶. In our patient data, *IL10RA* was indeed highly expressed in the effector cells (i.e., Th2 cells), which was even slightly upregulated at 8h following the ultra-rush antigen exposure in VAP. In the meantime, the antigen-specific response marker (*CD40LG*) was decreased in sorted Th2 cells at 8h immediately following AIT launch. This indicates that Bregs might be able to suppress allergen-specific Th2 responses during AIT in VAP. It is also worth to highlight that Bregs have been shown to only transiently express IL-10 following antigen stimulation and then differentiate into antibody-secreting cells⁹⁷, which might, at least partially, explain a rapid contraction of Bregs following the intensive repeated re-exposure with venom allergens during the first 8h period of ultra-rush AIT in VAP. Furthermore, Bregs can be reactivated upon reencounter with their cognate antigen/allergen in subsequent exposures, thereby providing long-term protection or immune tolerance during allergen re-exposures. Based on our and other Breg data in patients, we speculate that identifying and expanding allergen-specific Bregs could be applied as an allergen-specific cellular tolerance-priming therapy in other types of allergic diseases, where current AIT strategies are not sufficiently successful yet.

Reviewer #2 (Remarks to the Author):

The authors report the early immune changes during the initiation phase of insect-venom-specific immunotherapy vis-a-vis pollen allergen-specific immunotherapy using mass cytometry, bulk RNA-seq of sorted Th2 cells, MSD-based cytokine measurements, Kinome analysis, and clinical lab enumeration of immune cells in whole blood. Although the authors highlight the discovery of early-stage molecular and cellular switches in respective patient cohorts using multiomics approaches there are several shortcomings to this manuscript in its current format.

1) The rationale for the study is based on distinct durability of venom vs pollen-specific immunotherapy. It would thus be ideal to mention whether all the patients in VAP group showed

long-term tolerance as opposed to the lack of it in all/majority of those in PAP group. Table S1 should have details on clinical diagnostic matrix for VAP.

Reply: Thanks for pointing out this critical aspect. We now provided additional information regarding the clinical outcomes and diagnostics in **Supplementary Table 1** for both patient groups. For the diagnostic matrix of VAP, we have now added information related to anaphylaxis Mueller grades. We have now provided the AIT outcome data (**Supplementary Table 1**) either based on field sting reactions or negative skin test for VAP. In Luxembourg, we did not yet implement the sting challenge tests in the routine care of the hospitals. All the six reported patients showed no systematic reaction to the field sting(s), indicating a 100% AIT curative rate. Another VAP, who did not experience or report field stings yet, has shown a negative skin test, also strongly indicating a cure. We have now also integrated the VAP clinical outcome data into the sIgG4 and sIgE results, which are considered as immunological surrogate outcomes, in the new **Figure 6**.

For the PAP patients, we have now added the diagnostic information in **Supplementary Table 1** regarding seasonal allergic rhinoconjunctivitis (SARC), intermittent asthma (IAS), oral allergy syndrome (OAS) and eosinophilic esophagitis (EOE). We also added the disease duration before the administration of AIT. Our clinical team has used a scoring of patient symptom severity inspired by the European Academy of Allergy and Clinical Immunology (EAACI) Combined Symptom and Medication Score (CSMS) to retrospectively evaluate the PAP AIT outcomes. Eight out of 16 PAP (50%) were concluded “unimproved” while only seven showed an improvement and one with a strong improvement. This high heterogeneity in the AIT efficacy of PAP is sharply contrasted to the 100% curative rate in the VAP with confirmed clinical outcomes. We now also added them in **Supplementary Table 1** and integrated the AIT outcome information into the sIgG4 response Figure (**Supplementary Figure 9**) with different color codes, representing distinct outcomes. Interestingly, the sIgG4 response in PAP is very similar to that in VAP, where all PAP, independent of being clinically improved or not, showed an increase in the follow-up samples following AIT, although with a strong variation.

2) There are no starting hypotheses stated, and the manuscript appears as a loose, lengthy summary of all the comparisons made among various immune cell subsets rather than a focused, fluent, impactful report underlining clinically relevant/applicable findings.

Reply: We agree with the reviewer that there is no starting hypothesis in our manuscript, different from most of the classical studies. In fact, we applied a hypothesis-free and data-driven unbiased multi-omics strategy to this clinical study. To make this point crisp and clear, we now slightly rephrased a couple of words in Introduction (**line 103, page 4**), Results (**line 128, page 4**) and Discussion (**line 583, page 19**). Among others, it is clear that we would have no chance to identify the unexpected changes on those less-known CD4 Th1-Th17 and CD8 Tc1-Tc17 hybrid cell types, if we first made a predefined

hypothesis to only focus on other “mainstream” cell types (e.g., Th2 cells). As discussed by us and many others in different reviews or reports, predefined hypotheses have led to enormous critical discoveries in the last decades. However, whenever the particular immune cell type has already been pre-chosen, one can hardly discover something completely unexpected. One of our main incentives for performing such an unbiased, hypothesis-free analysis is the fact that >110 years after the first publication of AIT in patients, we are still lacking a deep understanding of the early molecular and cellular drivers of successful immune tolerance restoration to an allergen/antigen.

3) Allergen-specific immune responses are not investigated. Also, none of the observations have been validated via functional assays.

Reply: Thanks for pointing out this critical aspect. To address this comment of the reviewer and comment 1 of Reviewer 1, we have now performed 438 antigen-specific sIgE and 378 sIgG4 measurements in ~200 plasma or serum samples in an accredited clinical laboratory. Essentially, our results show that AIT induces sIgG4 responses in the follow-up samples (6-20 Months) in venom patients (VAP), although with a strong variation among individual patients (refer to **new Figure 6**). The ratios between sIgG4 and sIgE were also increased in the VAP follow-up samples in general, although varying substantially. The sIgG4 responses in PAP were very similar to that in VAP, where all PAP, independent of being clinically improved or not, showed an increase in the follow-up samples following AIT, although again with a strong variation (**Supplementary Figure 9**). Those varying IgG4 responses are all consistent with the existing literature and have been discussed by us accordingly in the main text.

While we did not perform the *in-vitro* Basophil Activation Test (BAT) as a functional readout for allergen reactivity of individual patients, we now re-analyzed our dataset for the frequency of CD63+ (the CD63 marker Ab was already included in our CyTOF panel) cells among basophils. As shown in **Supplementary Figure 4**, as expected, the overall frequency of CD63+ cells among basophils is generally very low directly *ex vivo*. Furthermore, we did not observe any significant difference between patient groups vs HC at different time points. We also did not find any significant AIT-induced effect at the analyzed time points vs baseline in both patient groups. One could envision this analysis as a proxy for an *ex-vivo* BAT assay.

The general low frequency of CD63+ cells among basophils in our *ex-vivo* dataset could be explained by the low concentration of injected specific allergen in our study even under ultra-rush up-dosing conditions in VAP. We used the maintenance dose of 100 µg of total venom extract (according to the AIT product sheet of Pharmedon-ALK), i.e., 100,000 ng of total extract. Assuming our body has ~5 liters of blood in general, this results in a final concentration of 20 ng/mL of extract (even without considering degradation, pharmacokinetics and tissue distribution). Considering the contained allergen in the venom extracts, for instance, Ves v 5 (which constitutes only approximately 5% of the total dry vespid venom), the concentration of this specific allergen in the blood would be only approximately **1 ng/mL**. However, as shown in Figure 7 of the published paper

(<https://journals.aai.org/jimmunol/article/184/9/5403/83318/Identification-Recombinant-Expression-and>) and many other papers, one might need to use at least **~10 ng/mL** of purified allergens to induce a clear positive BAT signal in cytometry. Therefore, one might need to consider the physiological relevance of the *in-vitro* BAT assay data with caution as a surrogate functional test regarding the clinical outcome in VAP and PAP. We now also discussed these data in line 218-223, page 7.

4) At certain instances, non-significant findings/trends are mentioned alongside the significant ones in the main text. Such non-significant indicators could be a part of supplemental data.

Reply: Thanks for pointing out this aspect. We now removed all the terms like “showing a trend be increased” or “showing a trend to be decreased” in the main text to only focus on those apparently significant results.

However, we would like to point out that in some places we still show non-significant results, especially those from the Pollen patient group, as a contrast to the significant VAP results. In any case, to allow the readers to directly compare the results, we often display VAP, PAP and HC data in the same panels to show the dynamic change patterns of one specific immune subset (in Main or Supplementary Figures).

5) The P values should be adjusted for multiple comparisons and the resulting q values (if satisfy the significance threshold) should be mentioned in brackets in the context of respective main text.

Reply: Thanks for pointing out this critical aspect. The assumption (pre-condition) to apply for multiple comparison correction (e.g., Benjamini-Hochberg or Bonferroni correction procedure) is that all the tests have to be **independent from each other** (please refer to the original article, <https://rss.onlinelibrary.wiley.com/doi/10.1111/j.2517-6161.1995.tb02031.x>; or the online statistics course document, under the section “Assumption”, [https://stats.libretexts.org/Bookshelves/Applied_Statistics/Biological_Statistics_\(McDonald\)/06%3A_Multiple_Tests/6.01%3A_Multiple_Comparisons](https://stats.libretexts.org/Bookshelves/Applied_Statistics/Biological_Statistics_(McDonald)/06%3A_Multiple_Tests/6.01%3A_Multiple_Comparisons)). However, this independence assumption does not meet for our volcano plots in different Main and Supplementary Figures because the relative frequency of different immune subsets is apparently dependent on many others. For instance, the relative frequency of classical monocyte, non-classical and intermediate monocytes is clearly depending on each other; the frequency of effector memory and central memory CD8 T cells also apparently affects each other. Therefore, it is not justified to apply any type of multiple-comparison correction method to those volcano plots, where >70 immune subsets that are not completely independent in terms of their relative frequency were analyzed.

But for the figure panels with dynamic curves (e.g., **Figure 3**), where we performed several comparisons during the time course of AIT, we now followed the reviewer's suggestion and applied the multiple-comparison correction with the corresponding q values (false discovery rates, FDR) directly in the corresponding Figure legends. Even in those cases, if the subset % is higher in VAP vs HC at 8h, there will be still a good chance that that subset % might be still higher in VAP at 24h. Therefore, although we directly added the q values in the corresponding Figure legends to give additional critical information for the readers, we do not feel confident to discuss the level of significance (or not) based on q-values in the main text, which might be misleading in some cases. This is very different from the classical GWAS analyses, for example, where one has to apply the multiple-group correction to many often-independent and significant SNPs. In our cases, one has to look at the results of multiple relevant immune subsets together to come to a fair conclusion.

6) Most sections under Results include a writeup, which is more in line with the Discussion. Also, at multiple instances, there is a reference to figures from preceding result sections which mars the flow.

Reply: Thanks for identifying the scenarios where we put a reference to Figures. We now systematically checked all the sentences where both references and Figures exist. We now changed the reference citation position to refer to the cell type(s), rather than our observations.

Moreover, following the suggestion regarding “writeup” under different sections of Results, we now removed the summary sentences under most of the sections or moved some of them to the Discussion section. For a few sections of Results, we still kept the writeup sentence(s) as we thought it was important to strengthen a particular message for the reader, if the Results and Discussion sections are not read at the same time.

7) Please rephrase "whole-genome RNA-sequencing" (Pg 5, line 132), "treatment-resulting peripheral immune responses" (Pg 5, line 144), "volcano-plot analysis", "outstanding diagnostic power" (Page 14, line 423), "clinical cure rates" (Page 15, Line 451).

Reply: Thanks for pointing out those wrongly phrased terms. We now rephrased them to “whole-genome RNA sequencing”, “resulting peripheral immune responses among VAP induced by AIT”, “volcano plot analysis” and “curative rates”, respectively. For the term “outstanding diagnostic power”, we now explained it by adding half a sentence “, a ROC value categorized as ‘outstanding’ as shown elsewhere (<https://pubmed.ncbi.nlm.nih.gov/20736804/>)”. That work was well accepted in the field of biomarker discovery (cited >3000 times). In that paper, the authors defined an AUC value >0.9 as the ‘outstanding’ category while the range between 0.8 and 0.9 as the ‘excellent’ category. For the term

“performed a volcano-plot analysis...”, we replaced it with “used the volcano plot to visualize (Figure 2C) the results of...”.

8) Notably, although CD8+ T cells...(Page 6, line 173). Please refer to PMID: 36333296, PMID: 32835695, and re-phrase.

Reply: Thanks for suggesting these two missing references. We now cited them in the corresponding sentence and rephrased the sentence accordingly.

Minor:

1) Citation style not in line with Nature Communications

Reply: Thanks for identifying this mistake. We now downloaded the newest Nature Communication reference style file and applied it to our manuscript reference citations.

Reviewer #4 (Remarks to the Author):

The manuscript by Pogorelov et al., titled "Multiomics approaches disclose very-early molecular and cellular switches during insect-venom allergen-specific immunotherapy," is an interesting read, providing novel insights into immune cell subtypes involved in venom or pollen allergen-induced allergies, as well as the mechanisms potentially driving the beneficial effects of AIT.

One immediate question is whether the two different time regimens for up dosing used in VAP vs PAP are comparable. The authors rightly address and acknowledge this issue.

The non-inflammatory IL-6 induction as a mechanism by which AIT may drive the switch from Th2/Tc2/insufficient CD8+ Treg towards Breg and the restoration of IL-6/APC equilibrium is truly intriguing. Including a schematic diagram depicting this process in the manuscript would be beneficial.

While I have no other major concerns, there are some minor points that could be addressed to improve the paper.

Reply: Thanks for your appreciation of our work. Based on the suggestion of this reviewer, we have now included a new schematic diagram as Figure 6D to summarize the various major observations. We also directly show it here in the response letter below. Of note, during the revision, we found the specific subset of monocytes (classical monocytes, cMono) showed a more significant correlation with IL-6 than

total APC in VAP following AIT. Furthermore, the contrast in the correlation degrees between cMono and IL-6 among the three groups (VAP, PAP and HC) at baseline is also stronger. Therefore, we now decided to replace total APC with cMono in **Figure 5 and 6**.

[Panel redacted]

Minor comments:

Line 87. The way this sentence reads now, suggests IgE levels are reduced while activation of mast cells and basophils is increased. Thus, please rephrase into e.g.,

This is demonstrated by a reduction in IgE levels (Michils et al. 2000, Galli et al. 2008), a decrease in the activation and degranulation of mast cells and basophils (Maintz et al. 2009, Akdis and Akdis 2014), and lower frequencies of allergen-specific Th2 cells (Wambre et al. 2017, Rudulier et al. 2019)

and type 2 innate lymphoid cells (ILCs) (Golebski et al. 2021).

Reply: Thanks for this constructive suggestion. In the revision, we now rephrased that sentence in a way as you suggested.

Fig 2A The UMAP visualization looks very blurry – could benefit from optimizing colours, contrast and resolution. Also the figure legend (annotation of cell subtypes within clusters) could be better designed in terms of size, font and colour.

Reply: Thank you for your feedback regarding the visualization (**Fig 2A** and **Fig 3C, 3H** in the next comment; in the revised version, **3H** is now re-ordered as **3I**) in our manuscript. We would like to address your concerns about the perceived blurriness of the plot from different angles.

Firstly, it appears there was a minor misunderstanding regarding the type of plot. These plots in our Figures are not UMAP plots, but rather a 2-dimensional embedding of the SOM clustering utilizing the function “EmbedSOM” within the “GigaSOM” package (<https://doi.org/10.12688/f1000research.21642.2>). We also directly mentioned the name ‘EmbedSOM’ in the Figure legend. In contrast to the UMAP, where downsampling is often required for a large number of cells and could risk the loss of some rare cell subpopulations, the EmbedSOM plot allows us to visualize every single data point (cell) in a 2-dimensional image. This image is constructed by transforming (rasterizing) each cell coordinate (x, y) into a pixel position. The blurriness you noted arises due to the extremely high density of data points. With around 200 million cells that have never been previously shown in one plot by others in single-cell based cytometry analyses, we have significantly more data points (i.e., cells) than pixels in the plot, which causes overlapping of different data points. Because of the overlapping issue, we previously applied a relatively high transparency (i.e., low alpha value) to each data point, which is the main cause of the perceived blurriness.

Nevertheless, after careful thinking and internal discussing during the revision, we decided to completely remove **Fig. 2A**. In addition to the aforementioned technical complexity, there are several other reasons urging us to remove **Fig. 2A**. First, **Fig. 2A** that only lists different immune cell types in a visually-attractive manner does not provide additional values to the manuscript as we already have **Supplementary Figure 1**, which have already shown all the analyzed subsets and the full gating strategy anyway. Second, in the revised **Fig 3C** and **3I**, we have now applied sub-clustering to further improve the resolution and highlighted cell types from our large-scale cytometry data analysis.

Line 156. The frequency of Th2 is not increased, but at a higher level in VAP across all timepoints

compared with HC

Reply: Thanks for identifying this mistake. We now rephrased it to “higher level” as suggested. We have now also rephrased the wording (using “higher” or “lower” or synonyms in the new version, rather than “increased” or “decreased” in the previous version) in other similar scenarios.

Fig 3C The visualization is again blurred. Please improve.

Reply: Please also refer to the response to this reviewer’s comment regarding “**Fig 2A**” above.

We now put efforts to reduce the perceived blurriness on both **Fig 3C** and **3H** (now called **3I** in the new version) by implementing different measures. First, we reduced the total data points (cells) by extracting only CD4 T cells from the total PBMC cells and performed sub-clustering only on CD4 T cells. In addition, we set the alpha values of highlighted subsets (e.g., Th2, CRTH2+ Th2 cells, Th17 and Th1-Th17 hybrids) to 1 (i.e., no transparency) while keeping the alpha values for the background at a low level. Furthermore, we reduced the raster image resolution which creates a denser area for the highlighted subset.

This shall help to illustrate the density and distribution of the data at a single cell level. We appreciate your understanding and hope this explanation clarifies the approach and rationale behind our large-scale visualization technique.

Line 211. The Th1-Th17 subset was not declined but at a lower level in PAP vs HC

Reply: Thanks for identifying this mistake. We now rephrased to “lower” as suggested in this and other places, whenever a similar scenario appears.

Fig5D please correct typo in “reponse to interleukin 6”

Reply: Thanks for identifying this typo. We now revised it to “Response to interleukin 6” directly in Figure 5.

All the reviewers have approved our revision last round and there is no more point-by-point response anymore.